# Ultraslow serotonin oscillations in the hippocampus delineate substates across NREM and waking

**Claire Cooper[1]\*, Daniel Parthier[1], Jeremie Sibille[1], John J Tukker[1,2], Nicolas Tritsch[3,4], Dietmar Schmitz[1,2,5,6]\***

[1]Charité-Universitätsmedizin Berlin, corporate member of Freie Universität Berlin and Humboldt-Universität zu Berlin, Neuroscience Research Center, Berlin, Germany; [2]German Center for Neurodegenerative Diseases (DZNE) Berlin, Berlin, Germany; [3]Neuroscience Institute, New York University Grossman School of Medicine, New York, United States; [4]Douglas Mental Health University Institute, Douglas Research Center, Montreal, Canada; [5]Charité-Universitätsmedizin Berlin, corporate member of Freie Universität Berlin and Humboldt-Universität Berlin, NeuroCure Cluster of Excellence, Berlin, Germany; [6]Humboldt-Universität zu Berlin, Bernstein Center for Computational Neuroscience, Berlin, Germany

**\*For correspondence:**
claire.cooper@charite.de (CC);
dschmitz-office@charite.de (DS)

**Competing interest:** The authors declare that no competing interests exist.

## eLife Assessment

This **important** study demonstrates that slow fluctuations in serotonin release during wakefulness and non-REM sleep correspond to periods of heightened arousal or enhanced offline information processing. The evidence supporting this claim is **convincing**, and the methodology is robust and broadly applicable, likely to benefit many researchers in the field. This work will be of significant interest to neuroscientists studying sleep, memory, and neuromodulation.

**Abstract** Beyond the vast array of functional roles attributed to serotonin (5-HT) in the brain, changes in 5-HT levels have been shown to accompany changes in behavioral states, including WAKE, NREM, and REM sleep. Whether 5-HT dynamics at shorter time scales can be seen to delineate substates within these larger brain states remains an open question. Here, we performed simultaneous recordings of extracellular 5-HT using a recently developed G-Protein-Coupled Receptor-Activation-Based 5-HT sensor (GRAB5-HT3.0) and local field potential in the hippocampal CA1 of mice, which revealed the presence of prominent ultraslow (<0.05 Hz) 5-HT oscillations both during NREM and WAKE states. Interestingly, the phase of these ultraslow 5-HT oscillations was found to distinguish substates both within and across larger behavioral states. Hippocampal ripples occurred preferentially on the falling phase of ultraslow 5-HT oscillations during both NREM and WAKE, with higher power ripples concentrating near the peak specifically during NREM. By contrast, hippocampal–cortical coherence was strongest, and microarousals and intracranial EMG peaks were most prevalent during the rising phase in both wake and NREM. Overall, ultraslow 5-HT oscillations delineate substates within the larger behavioral states of NREM and WAKE, thus potentially temporally segregating internal memory consolidation processes from arousal-related functions.

## Introduction

The impact of the outside world on neural activity is highly dynamic and dependent on the state of the brain. During waking behavior, sensory stimuli are actively processed by the brain and shape ongoing brain activity, whereas during sleep, the impact of such external stimuli is reduced in favor of internally generated rhythms. Transition among behavioral states is accompanied by changes in the extracellular levels of neuromodulators. One such neuromodulator, serotonin (5-HT), shows clear state-dependent changes in activity, with the firing of 5-HT neurons in the brainstem being highest during waking, intermediate during NREM, and lowest during REM states (*McGinty and Harper, 1976*; *Trulson and Jacobs, 1979*). Furthermore, changes in 5-HT levels have been causally linked to brain state changes, though some controversy over the direction of such changes remains. While some studies suggest a wake-promoting role for 5-HT, others propose that 5-HT increases sleep drive over the course of waking (*Ursin, 2008*; *Monti et al., 2008*). In either case, state-dependent changes in 5-HT levels can be seen to reorganize brain networks in response to ongoing functional demands.

Beyond traditional brain states, recent attention has been drawn to the existence of substates within these larger brain states. NREM sleep, for example, has been shown to contain periods of high and low arousal (*Lecci et al., 2017*; *Osorio-Forero et al., 2021*). In these studies, high arousal substates were associated with higher heart rate and sensitivity to auditory stimuli, while low arousal substates contained more hippocampal ripples and sleep spindles. Therefore, substates in NREM may mediate the balance between processing external stimuli and carrying out internal brain processes, such as memory consolidation. Importantly, in both studies, these substates were delineated by the phase of ultraslow oscillations (<0.1 Hz) of sigma power and noradrenaline levels, respectively. While NREM substates have not yet been examined in relation to 5-HT levels, ultraslow oscillations have been observed in population activity in the dorsal raphe nucleus (DRN) (*Kato et al., 2022*; *Mlinar et al., 2016*), as well as in extracellular 5-HT levels in the hippocampal dentate gyrus (*Turi et al., 2024*) during NREM, suggesting that 5-HT may also distinguish pro-arousal and pro-memory substates.

5-HT is a key modulator for many brain functions, which is reflected by the highly extensive projections of serotonergic fibers throughout the mammalian brain. Especially dense are the connections from the midbrain raphe nuclei, the source of 5-HT to the hippocampus, a region important for memory processing (*Jacobs and Azmitia, 1992*). Hallmarks of the hippocampus, ripples are transient fast oscillations (120–250 Hz) observed in the local field potential (LFP) and have been shown to functionally underlie memory consolidation and replay (*Buzsáki, 2015*). While the contribution of 5-HT to memory processing remains unclear, with different studies supporting facilitating versus suppressing roles (*Meneses, 2015*; *Coray and Quednow, 2022*, *Teixeira et al., 2018*; *Zhang et al., 2013*; *Zhang and Stackman, 2015*; *Zhang et al., 2016*), the three studies examining the effect of 5-HT modulation on ripples all found a suppressive effect (*Wang et al., 2015*; *ul Haq et al., 2016*; *Shiozaki et al., 2023*). However, interpreting these studies is made difficult by the methods used to manipulate 5-HT levels, namely optogenetic activation of 5-HT neurons and pharmacologic interventions at high doses. Simultaneous brain-wide increases of 5-HT levels resulting from optogenetic raphe stimulation and pharmacological interventions via systemic injections have the potential to activate 5-HT sub-systems which are not naturally active together, such as the reward- and movement initiation-activated serotonergic fibers described in the dorsal hippocampus (*Luchetti et al., 2020*). Further rendering these studies hard to interpret is the question of physiologically plausible dose, as biphasic dose-dependent effects of 5-HT have been described (*Calabrese, 2001*).

To bypass such constraints, in the present study we utilized the recently developed G-Protein-Coupled Receptor-Activation-Based (GRAB) 5-HT sensor which allows for the measurement of physiological changes in local extracellular 5-HT concentrations with high spatial and temporal resolution (*Deng et al., 2024*). Alongside 5-HT levels, we recorded hippocampal activity with silicon probes in order to examine potential correlations between local 5-HT fluctuations and behavioral states and substates in the dorsal CA1 in freely moving mice. After simultaneous fiber photometry and LFP recordings, we could identify substates of NREM and WAKE delineated by different phases of ultraslow 5-HT oscillations. These substates roughly corresponded to periods of higher and lower arousal, with lower arousal associated with the preferential occurrence of ripples, and higher arousal with the occurrence of microarousals (MAs) as well as peaks in the intracranial EMG (icEMG, see *Methods: State scoring*) and hippocampal–cortical coherence.

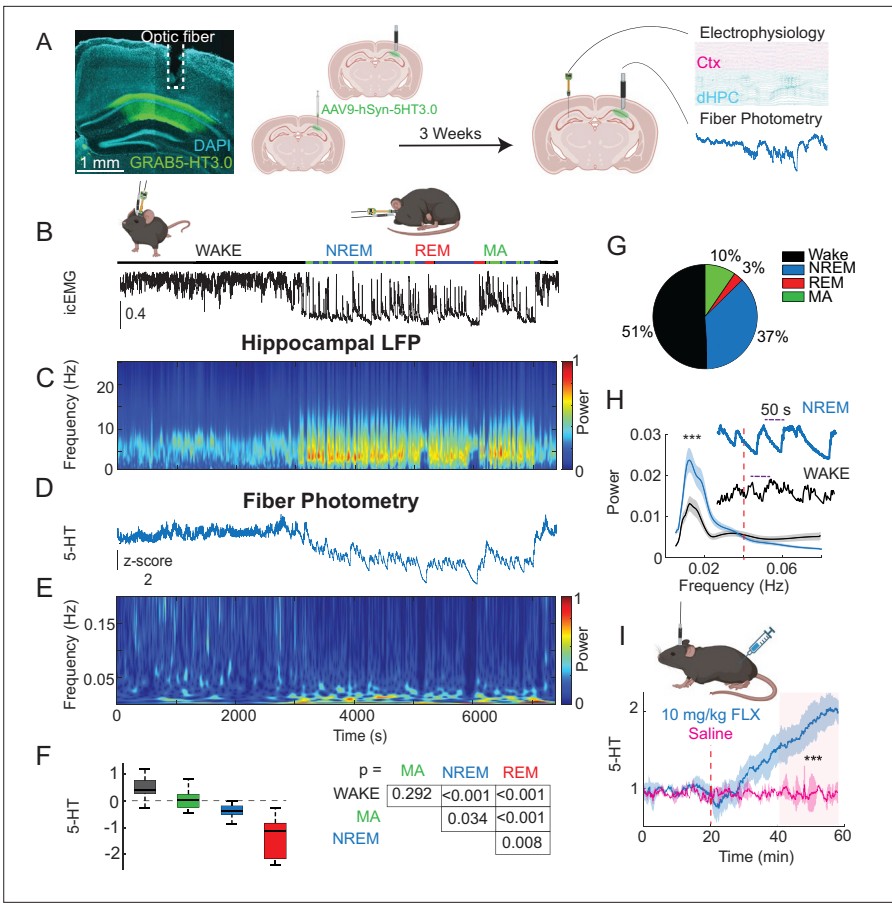

**Figure 1.** 5-HT levels exhibit ultraslow oscillations during NREM and WAKE. (**A**) Histology and experimental protocol. Left: expression of GRAB5-HT3.0 sensor (in green) in dorsal CA1 with optic fiber track above. Right: methodology for dual implantation surgeries. AAV9-hSyn-5HT3.0 was first injected into the right dorsal CA1. In the same surgery, an optic fiber was implanted above the injection site. After 3 weeks of viral expression, a silicon probe was implanted above the left dorsal CA1. Simultaneous recording of the GRAB5-HT3.0 sensor activity (fiber photometry) and electrophysiology was performed. (**B–E**) Example dual fiber photometry–electrophysiology recording with times shown in (**E, B**). Labeled sleep states resulting from automated sleep-scoring above an intracranial EMG trace. (**C**) Spectrogram (Stockwell transform) showing normalized power of a hippocampal local field potential (LFP) channel during awake and sleep states. (**D**) Z-scored 5-HT trace. (**E**) Spectrogram (Stockwell transform) of the 5-HT trace shown in (**D, F**) Left: mean 5-HT level by state, across all experiments (total $n = $ 6 mice, 12 recording sessions of 1.5–3 hr). Error bars represent standard error of the mean (SEM). Right: p-values from a multiple comparisons test applied after fitting a Bayesian general linear mixed model (GLMM) to the data. (**G**) Pie chart showing proportion of time spent in different behavioral states, averaged across all experiments. (**H**) Top: examples of ultraslow 5-HT oscillations in NREM and WAKE. Bottom: power spectrum of 5-HT signals in WAKE versus NREM sleep. Frequencies were separated into two groups (>0.04 and <0.04 Hz), and a GLMM was fitted to the data. p-values shown were derived from a post hoc multiple comparisons test on the fitted model. Results indicated higher power in the lower half of frequencies (<0.04 Hz) for both states ($p < 0.0001$), as well as a significant difference in the lower frequencies between NREM and WAKE ($p < 0.0001$), with an NREM/WAKE ratio of 1.21 ± 0.028. (**I**) Control fluoxetine and saline injection experiments. A significant difference between the post-injection period of saline- and fluoxetine-injected animals (shaded in red) was observed (Wilcoxon ranked-sum test, $p < 0.001$, $n = 3$ mice). Asterisks indicate statistical significance: * = $p < 0.05$, ** = $p < 0.01$, *** = $p < 0.001$.

The online version of this article includes the following figure supplement(s) for figure 1:

**Figure supplement 1.** Waveform of ultraslow 5-HT oscillation is asymmetric.

# Results

In order to simultaneously record local 5-HT levels and electrophysiological signals, we first injected mice with a virus encoding the GRAB5-HT3.0 sensor (AAV9-hSyn-5HT3.0) in the right dorsal CA1 (*Deng et al., 2024*). Simultaneous fiber photometry and electrophysiology recordings were achieved by implanting an optic fiber (400 μm diameter) above the injection site (*Figure 1A*) and a silicon probe in the left dorsal CA1, at the same anterior–posterior coordinates as the site of viral injection (see *Methods*).

To verify that the sensor reports changes in local 5-HT levels, we treated a subset of mice (*n* = 3) with fluoxetine (10 mg/kg), an SSRI known to acutely increase extracellular 5-HT levels in the dorsal hippocampus (*Imoto et al., 2015*). Compared to saline, fluoxetine significantly elevated fluorescence recorded by fiber photometry, confirming the sensitivity of the GRAB-5HT3.0 sensor to endogenous levels of 5-HT in the hippocampus of mice (*Figure 1I*).

In order to examine the relationship between 5-HT signals and hippocampal activity across brain states, we simultaneously recorded hippocampal GRAB5-HT3.0 fluorescence and LFPs in the home cage during normal behavior, which included both waking and sleeping bouts. Twelve recording sessions were conducted using six mice (1–3 sessions per mouse), all of which were included in subsequent analysis and statistical testing. Automated sleep-scoring (*Watson et al., 2016*) was performed on the LFP data, showing the occurrence of different behavioral states, namely WAKE, non-REM sleep (NREM), REM sleep, and MAs, with different frequencies throughout the recording sessions (*Figure 1G*). Consistent with prior reports in the hippocampus (*Park et al., 1999*), 5-HT levels were highest during WAKE, intermediate during NREM, and lowest during REM (*Figure 1F*). During MAs, 5-HT levels were on average in between WAKE and NREM levels, but statistically indistinguishable from WAKE. Most strikingly, we observed prominent ultraslow (<0.06 Hz) oscillations of 5-HT levels during NREM (*Figure 1D, E*). These ultraslow oscillations were reflected as a clear peak in the power spectrum at ~0.015 Hz, with significantly greater power in the <0.04 Hz range than in the >0.04 Hz range for both states (*Figure 1H*). Similar ultraslow oscillations were observed during WAKE, though with significantly less power than NREM ultraslow oscillations (NREM/WAKE = 1.21 ± 0.028).

The waveform of these ultraslow 5-HT oscillations was shown to be asymmetric with respect to both slope (*Figure 1—figure supplement 1A, B*) and length (*Figure 1—figure supplement 1C, D*), with a steeper and shorter rising phase and a flatter and longer falling phase. Although the GRAB5HT3.0 sensor we use exhibits a similar kinetic asymmetry, with a shorter rise time (0.25 s) than decay time (1.39 s) (*Deng et al., 2024*), we consider it unlikely to account for the asymmetry observed in ultraslow 5-HT oscillations. For one, the 5-HT signal clearly contains reductions in 5-HT levels that are much faster than the descending phase of the ultraslow oscillation. Furthermore, the finding that a subset of dorsal raphe neurons firing at ultraslow frequencies displays clear asymmetry in rising time versus decay suggests that the asymmetry we observe in our data could be due to neural activity rather than temporal smoothing by the sensor (*Mlinar et al., 2016*). In this same direction, another study found similar asymmetry in extracellular 5-HT levels measured with fast scan cyclic voltammetry, a technique with greater temporal resolution than our sensor, after single pulse stimulation of serotonin fibers (*Bunin and Wightman, 1998*).

Next, we sought to investigate whether substates could be defined relative to the phase of these ultraslow 5-HT oscillations, as previously described for sigma power and noradrenaline oscillations (*Lecci et al., 2017*; *Osorio-Forero et al., 2021*). The stronger ultraslow oscillations of 5-HT during NREM as compared to WAKE states led us to hypothesize a stronger coupling between ultraslow 5-HT oscillations and hippocampal activity during NREM than during WAKE.

## 5-HT and ripples

First, we looked at the relationship between ultraslow 5-HT oscillations and hippocampal ripples, the electrophysiological signature of memory consolidation. Given the noted shortcomings of the standard spectral filter-based methods for detecting ripples (*Liu et al., 2022*), and the recent surge of papers proposing alternative detection algorithms (*Watanabe et al., 2021*; *Hagen et al., 2021*; *Navas-Olive et al., 2022*; *Sebastian et al., 2023*), we chose to detect ripples with a custom convolutional neural network (CNN) model (see *Methods*). 400 ms segments of eight LFP channels, including four neocortical and four hippocampal channels, served as input to the model. After training and validation, the best-performing model consisted of four 2D convolutional blocks followed by two dense

layers, outputting a 400-ms ripple probability vector (*Figure 2A*). This output vector was thresholded to detect ripples. Notably, the model was able to successfully distinguish ripples from non-ripple uniformly propagating fast oscillations and movement-related noise (*Figure 2B, C*). Importantly, the features of detected events fell within the bounds expected for hippocampal ripples, with ripple duration and *z*-scored power showing a log-normal distribution (*Levenstein et al., 2019*; *Figure 2D*).

In comparing ripple occurrence to the 5-HT signal, we noticed that both peaks of power in the ripple band (120–250 Hz) (*Figure 2E*), a measure independent from ripple detection, and detected ripples (*Figure 2F*) tended to occur on the falling phase of slow 5-HT oscillations. In order to get a better sense of the timing between the 5-HT signal and ripple occurrence, we extracted ripple clusters, which were defined as groups of 10 or more ripples with an inter-ripple interval (IRI) of 3 s or less (*Figure 2G*). These parameters were empirically observed to capture the clusters of ripples of various lengths occurring during the falling phase of 5-HT oscillations and to exclude the less numerous, non-clustered ripples occurring on the rising phase. Having defined ripple clusters, we were then able to isolate the first and last ripples of each cluster. When considering all ripples, before cluster extraction, the average ripple was seen to occur on the falling phase of the 5-HT oscillation, in both NREM and WAKE (*Figure 2H1, H2*, first column). The first ripple of ripple clusters consistently occurred shortly after the peak in 5-HT in NREM, and at the peak in WAKE, while the last ripple occurred at the trough in both NREM and WAKE (*Figure 2H1, H2*, columns 2 and 3). In summary, ripples were seen to span the length of the falling phase of ultraslow 5-HT oscillations in both NREM and WAKE, though this relationship was generally stronger in NREM.

To further probe the relationship between ripple occurrence and 5-HT, we examined when ripples preferentially occur in relation to the phase of ultraslow 5-HT oscillations. We first looked at the distribution of IRIs along different phases of ultraslow 5-HT oscillations measured in the 0.01–0.06 Hz frequency band (*Figure 3A–D*), which showed the strongest temporal coupling to ripples compared to other slow oscillation frequencies (*Figure 3—figure supplement 1*). There, consistent with what we observed in the cluster extraction data, we found lower IRIs along the falling phase (–180°:0°), and higher IRIs along the rising phase (0°:180°) (*Figure 3B, D*). These differences in IRIs between rising and falling phases were statistically significant for both NREM and WAKE (*Figure 3C*). As observed previously, the phase preference of NREM ripples was stronger than that of WAKE ripples ($p < 0.0001$), though both were prominent.

Next, we calculated the ultraslow 5-HT phases at which individual ripples occurred during both NREM and WAKE (3E–F), where we again observed a significant falling phase preference in both states, albeit again stronger in NREM than WAKE ($p < 0.0001$), confirmed by a general linear mixed model (GLMM) where phase preference was modeled as a binary outcome (rising vs. falling) (*Figure 3F2 and F4*). Despite more session-level variance in the preferred phase angles of WAKE ripples (*Figure 3—figure supplement 1*), the overall mean phase angles of NREM and WAKE ripples were very similar, at 101.1° and 99.6°, respectively, reflecting a very similar distribution of ripples along 5-HT ultraslow oscillations across states (*Figure 3E3*). This result appears to be at odds with the mean ripple timing relative to 5-HT in NREM shown in *Figure 2H1*, column 1, where the mean ripple falls closer to the 5-HT trough. However, this apparent discrepancy can be explained by the method used to determine the mean ripple timing. In *Figure 2H*, the analysis is agnostic to the frequency of the underlying serotonin oscillation, meaning data is being averaged in segments with different oscillation periods. This leads to some imprecision in the estimation of the exact location along the falling phase where the mean ripple lies. In *Figure 2H*, columns 2 and 3, this is less of a problem, because the serotonin segments are 'anchored' by the peak and trough, respectively, as the onsets and offsets of ripple clusters have a clear preference for these time points. Therefore, the fact that the length of the falling phase varies is less important. In *Figures 3F2–F4*, the preferred phase for each ripple is calculated on data filtered in the ultraslow frequency band (0.01–0.06 Hz), so noise from faster or slower rhythms is not included. Phase estimation then accounts for the variability in oscillation period, which within ultraslow frequencies can range from 16.7 to 100 s. In summary, the phase analysis likely provides a more precise estimate of the preferred location of ripples within serotonin ultraslow oscillations. However, given that the distribution does not contain a significant peak but is closer to a uniform distribution within the falling phase, the mean ripple is less meaningful than the general distribution of ripples along the rising and falling phase.

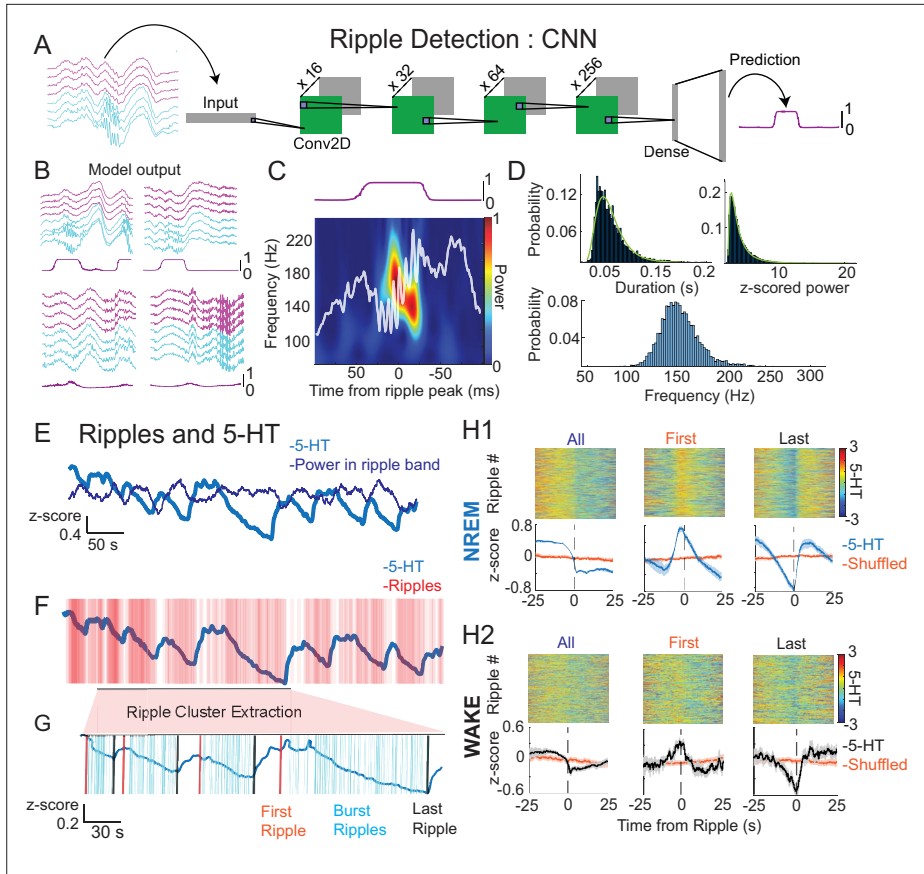

**Figure 2.** Ripples occur time-locked to ultraslow 5-HT oscillations. (**A**) Schematic showing the convolutional neural network used for ripple detection. 8-channel × 400-ms local field potential (LFP) chunks were used as input. The bottom four channels (cyan) were taken from the dorsal CA1 and contained ripples, and the top four channels (magenta) were chosen from a non-adjacent part of the neocortex above the dorsal CA1. The model consisted of four convolution blocks (Conv2d), each block comprising a 2D convolutional layer, a ReLU activation function, and batch normalization. Two dense layers with dropout and batch normalization (Dense) followed and produced the final output, a 400-ms vector with values between 0 and 1, indicating the probability of a ripple occurring during the course of the input chunk. (**B**) Example model output given the four LFP chunk inputs shown. First row: true positives. Second row: fast oscillations and movement artifacts not detected as ripples by the model. (**C**) Spectrogram from a ripple detected by the model, 0–1 normalized. (**D**) Characteristics of detected ripples. Ripples from all experiments were included, and probability distributions are shown. Top left: distribution of duration. Top right: distribution of $z$-score normalized ripple power. Bottom: distribution of ripple frequency. Ripple duration and normalized ripple power follow a log-normal distribution (duration: $X^2$ (df = 7, $N$ = 49,458) = 1.398e+03, p < 0.0001, normalized ripple power: $X^2$ (df = 7, $N$ = 49,458) = 422.1862, p < 0.0001). (**E**) Example 5-HT trace and computed power in the ripple band (120–250 Hz). (**F**) Same example 5-HT trace and individual detected ripples. (**G**) Example of ripple cluster extraction. Ripple clusters were defined as having a minimum of 10 ripple events and an inter-ripple interval of less than 3 s. Note the few ripples occurring during the rising phase of 5-HT ultraslow oscillations in (**F**) are excluded from extracted ripple clusters in (**G**). From these ripple clusters, the first (orange) and last (black) ripples in a cluster were extracted. (**H1, H2**) Ripple-triggered 5-HT in NREM (H1) and WAKE (H2) states. The first rows of H1 and H2 show all 50 s 5-HT segments centered around the ripple peak for different combinations of ripples (columns). In the first column, all ripples in the given state were included. The second and third columns used only the first or last ripple in extracted ripple clusters, respectively. The second rows of H1 and H2 show the mean ripple-triggered 5-HT traces (blue) and randomly shifted traces (orange) for each group of ripples. The orange traces were obtained by randomly shifting the ripple times for each condition and averaging the resulting 5-HT 50 s segments centered around those shifted times.

The online version of this article includes the following figure supplement(s) for figure 2:

**Figure supplement 1.** Ripples missed by the model do not affect ripple distribution along ultraslow 5-HT oscillations.

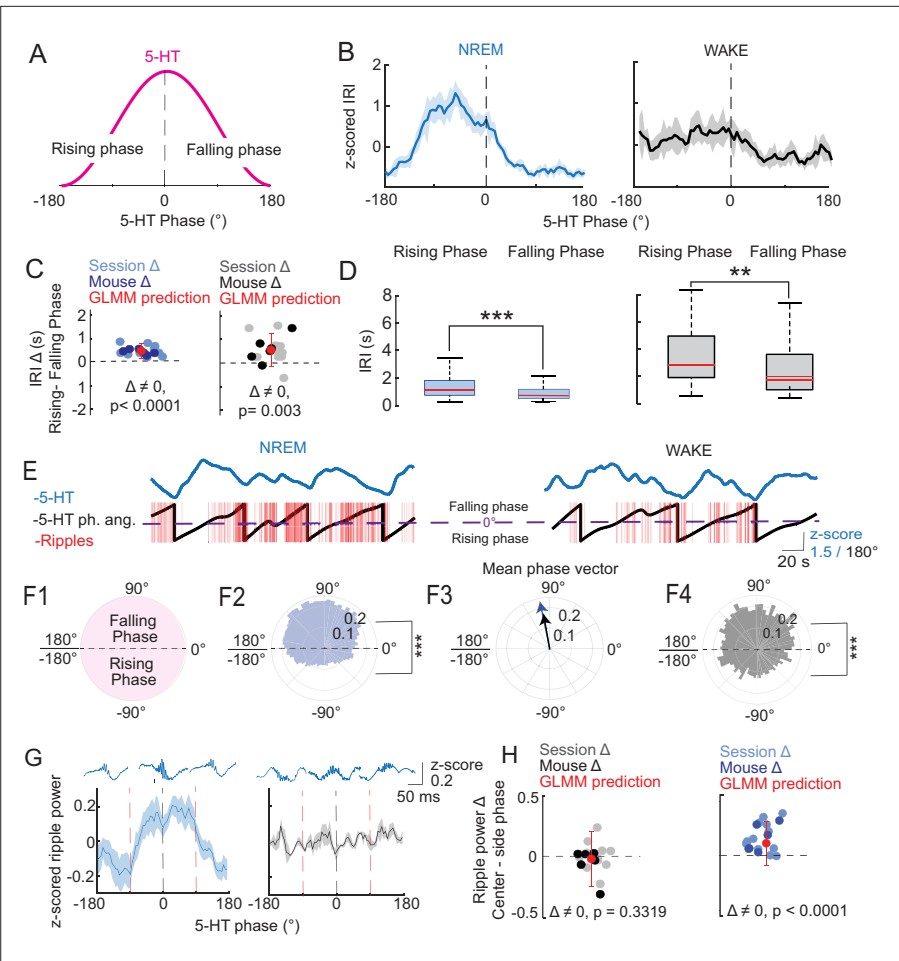

**Figure 3.** Ripple occurrence and power vary by the phase of ultraslow 5-HT oscillations. (**A**) Schematic showing one period of a slow 5-HT oscillation. The rising phase of the oscillation occurs from −180° to 0°, and the falling phase occurs from 0° to 180°. (**B**) Mean z-scored inter-ripple interval (IRI) by 5-HT phase angle during NREM (left) and WAKE (right). (**C**) Mean rising phase IRI − mean falling phase IRI, plotted by session and mouse level in WAKE (left) and NREM (right). Red point with error bar indicates predicted difference and confidence interval after fitting a general linear mixed model (GLMM) to the data. p-values shown were derived from a post hoc multiple comparisons test on the fitted model (*n* = 6 mice, 12 sessions). (**D**) Box plots showing IRI in rising versus falling phase in NREM (left) and WAKE (right). Significance levels shown reflect results of statistical analysis in (**C,E**). Asterisks indicate statistical significance: * = p < 0.05, ** = p < 0.01, *** = p < 0.001. Error bars represent SEM. Example 5-HT trace (top) and corresponding 5-HT phase angles and ripples (bottom) for NREM (left) and WAKE (right). The peak of the ultraslow oscillation (0°) is indicated by the dashed purple line. (**F1**) Schematic polar plot showing one period for a slow 5-HT oscillation. The falling phase of the oscillation occurs from 0° to 180°, and the rising phase occurs from −180° to 0°. (**F2**) Phase of all NREM ripples relative to the ultraslow 5-HT oscillation. A significant (p < 0.0001) bias toward the falling phase was found after fitting a GLMM to the data with a binomial link function. (**F3**) Mean phase vector of NREM and WAKE ripples. A significant (p < 0.0001) bias toward the falling phase was found after fitting a GLMM to the data with a binomial link function. Asterisks indicate statistical significance: * = p < 0.05, ** = p < 0.01, *** = p < 0.001. (**F4**) Phase of all WAKE ripples relative to the ultraslow 5-HT oscillation. (**G**) Z-scored ripple power by 5-HT phase angle during NREM (left) and WAKE (right). Red vertical dashed lines delineate analyzed phase segments: 'center' (−90° to 90°) versus 'side' (−180° to −90° and 90° to 180°). Representative ripples from each phase grouping are shown above. (**H**) Mean center phase ripple power − mean side phase ripple power, plotted by session and mouse level in WAKE (left) and NREM (right). Red point with error bar indicates predicted difference and confidence interval after fitting a GLMM to the data. p-values shown were derived from a post hoc multiple comparisons test on the fitted model.

The online version of this article includes the following figure supplement(s) for figure 3:

**Figure supplement 1.** Ripples couple most strongly to ultraslow (0.01–0.06 Hz) oscillations of serotonin.

**Figure supplement 2.** Ripple frequency, but not duration, is shaped by ultraslow 5-HT oscillations.

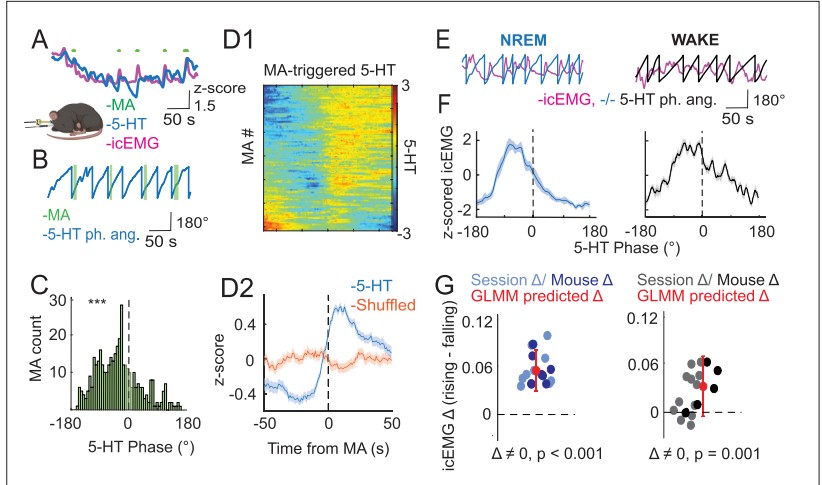

**Figure 4.** EMG and microarousals (MAs) vary by the phase of ultraslow 5-HT oscillations. (**A–D2**) Relationship between MA occurrence and the phase of slow 5-HT oscillations. (**A**) Example trace showing 5-HT, EMG, and MAs during an NREM bout. (**B**) Example trace showing extracted 5-HT phase angle and MAs. (**C**). MA occurrence according to 5-HT phase angle. A significant (p < 0.0001) bias toward the falling phase was found after fitting a general linear mixed model (GLMM) to the data with a binomial link function. Asterisks indicate statistical significance: * = p < 0.05, ** = p < 0.01, *** = p < 0.001. (**D1**) MA-triggered 5-HT across all MA events. (**D2**) Mean MA-triggered 5-HT trace (blue) plotted with mean of randomly shifted 5-HT trace (orange). The orange trace was derived by randomly shifting all MA times and averaging the resulting 5-HT segments around those shifted times. (**E–G**) Relationship between the EMG signal and phase of slow 5-HT oscillations. (**E**) Example traces showing extracted 5-HT phase angle and the EMG signal during NREM (left, blue) and WAKE (right, black) states. (**F**) Mean z-scored EMG signal by 5-HT phase angle during NREM and WAKE states. (**G**) Mean rising phase EMG − mean falling phase EMG, plotted by session and mouse level. Red point with error bar indicates predicted difference and confidence interval after fitting a GLMM to the data. p-values shown were derived from a post hoc multiple comparisons test on the fitted model.

Finally, we wondered whether ripples with different features were preferentially distributed according to the phase of ultraslow 5-HT oscillations. To this end, we first calculated the peak power for each detected ripple. Unlike what we previously observed with respect to ripple occurrence, the power of ripples in NREM showed a significant preference for the peak of ultraslow 5-HT oscillations, whereas no preference was observed in WAKE (*Figure 3F, G*). In summary, while ripples tend to show a preference for the falling phase of ultraslow 5-HT oscillations, stronger ripples tend to be statistically more likely near the peak of 5-HT in NREM, but not WAKE. We then performed the same analysis, looking this time at ripple duration and frequency with respect to ultraslow 5-HT phase. While ripple frequency was highest during the falling phase of ultraslow 5-HT oscillations (*Figure 3—figure supplement 2A, B*), ripple duration showed no clear phase preference (*Figure 3—figure supplement 2C, D*).

## 5-HT and MAs

Given that ripples, the electrophysiological signature of memory consolidation, were shown to constitute one substate occurring during the falling phase of ultraslow 5-HT oscillations, we next looked for signs of arousal-associated substates, potentially occurring at different phases of the ultraslow 5-HT oscillation. To this end, we first considered the occurrence of MAs relative to 5-HT, given that MAs themselves constitute periods of heightened arousal within NREM (*Halász, 1998*, *Watson et al., 2016*). MAs were observed together with peaks in the icEMG during NREM, which appeared to be time-locked to 5-HT ultraslow oscillations (*Figure 4A*). On average, MAs occurred shortly before the peak of 5-HT on the rising phase (*Figure 4D1, D2*). The same trend was observed when looking at the ultraslow 5-HT oscillation phase at which individual MAs occurred, where a significant preference for the falling phase was identified (*Figure 4C*), with the added information that MAs were generally much more likely on the rising phase than the falling phase. Therefore, not only do MAs themselves

define periods of arousal within NREM, but their occurrence is biased by the phase of the ultraslow 5-HT oscillation, which designates periods when such arousals can occur.

### 5-HT and icEMG

As MAs are only observed during NREM, we could not perform the same analysis on WAKE data. However, given that MAs are accompanied by peaks in the icEMG, we next examined whether the icEMG itself is time-locked to the phase of ultraslow 5-HT oscillations across states. Not surprisingly given the MA results, icEMG peaks during NREM occurred preferentially during the rising phase of ultraslow 5-HT oscillations (*Figure 4E, F*, left). Interestingly, icEMG peaks during WAKE were also observed preferentially in the rising phase (*Figure 4E, F*, right). The difference in icEMG between the rising and falling phases was shown to be significant in both states, with the effect in NREM being stronger than that in WAKE (*Figure 4G*). Therefore, the rising phase of ultraslow 5-HT oscillations can be seen to constitute a substate of heightened arousal, both in terms of MA occurrence during NREM, and the icEMG itself during both NREM and WAKE states.

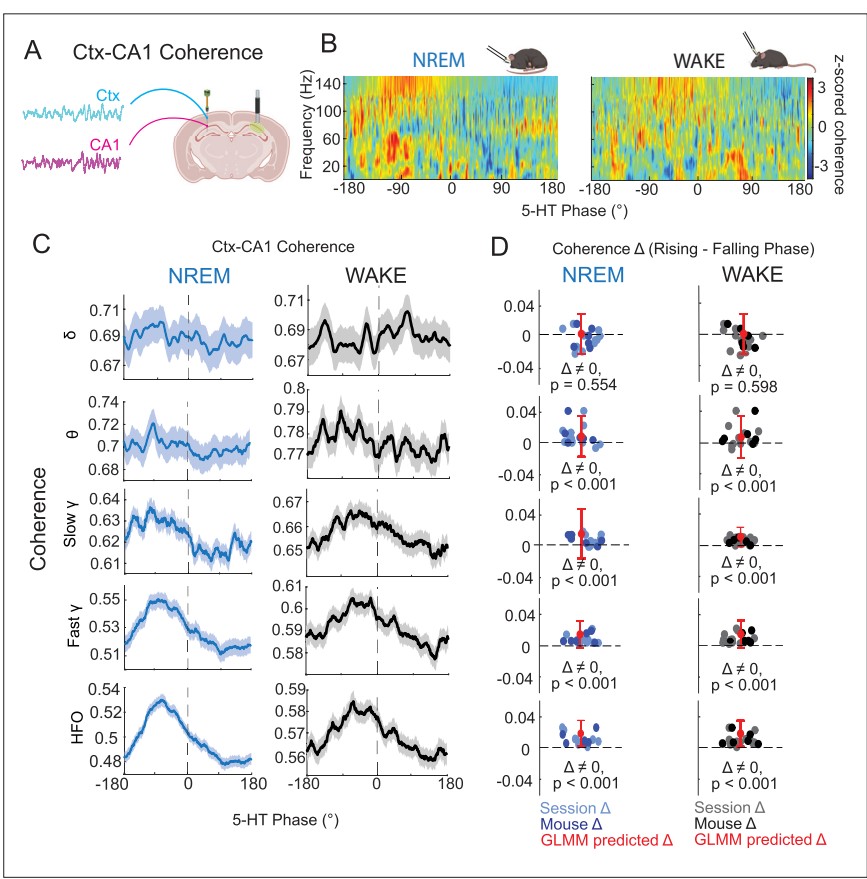

**Figure 5.** Coherence varies by the phase of ultraslow 5-HT oscillations. (**A**) Schematic showing representative hippocampal and cortical traces used for coherence calculations. (**B**) Mean z-scored hippocampal–cortical coherence by frequency for NREM (left) and WAKE (right). (**C**) Mean coherence by 5-HT phase angle for delta (1–5 Hz), theta (6–10 Hz), slow gamma (30–60 Hz), fast gamma (60–100 Hz), and high-frequency oscillation (HFO, 100–150 Hz) bands in NREM (left column) and WAKE (right column). (**D**) Mean rising phase coherence − mean falling phase coherence, plotted by session and mouse level for different frequency bands (rows) and states (columns). Red point with error bar indicates predicted difference and confidence interval after fitting a general linear mixed model (GLMM) to the data. p-values shown were derived from a post hoc multiple comparisons test on the fitted model.

The online version of this article includes the following figure supplement(s) for figure 5:

**Figure supplement 1.** Power in different frequency bands is timed by ultraslow 5-HT oscillations.

**Figure supplement 2.** Units show preference for different phases of ultraslow serotonin oscillations.

## 5-HT and hippocampal–cortical coherence

Coherence is a measure of synchrony between two brain areas thought to underlie neural communication (*Fries, 2005*). Changes in coherence, including hippocampal–cortical coherence, have been found to track changes in arousal, both across and within brain states (*Bastuji et al., 2021*; *Klaver et al., 2023*; *Cantero et al., 2004*). We therefore examined hippocampal–cortical coherence in relation to the phase of ongoing ultraslow 5-HT oscillations (*Figure 5A*). After computing coherence between pairs of hippocampal and cortical channels in both NREM and WAKE, we observed that in certain frequency bands, coherence was higher in the rising phase of ultraslow 5-HT oscillations than in the falling phase, in both states (*Figure 5B*). In order to more closely examine this trend, we looked at coherence by ultraslow 5-HT oscillation phase for each frequency band individually (*Figure 5C*). We found significant differences in coherence by ultraslow 5-HT phase in theta, slow gamma, fast gamma, and high-frequency oscillation bands, but not in the delta band both in NREM and WAKE (*Figure 5D*). Thus, inter-areal neural communication seems to be gated by the phase of ultraslow 5-HT oscillations, whereby during the rising phase, coherence is higher and such communication is favored.

Given these results, one could posit that the higher hippocampal–intracortical coherence we observe in the rising phase is due to a lack of activity during this time period, as no activity is easier to 'synchronize' than bouts of high activity. In fact, this potential relative silence in the rising phase would be in line with the observation that increases in DRN population activity are associated with decreased wideband power during sleep (*Kato et al., 2022*). To address this question, we looked into both the power of different hippocampal oscillations and unit activity relative to ultraslow 5-HT oscillations. While increased power during the falling phase was found for all frequency bands during NREM, during WAKE, gamma band power deviated from this pattern (*Figure 5—figure supplement 1A–C*). Specifically, low gamma showed no phase preference, whereas high gamma power was highest around the peak of ultraslow 5-HT oscillations. As hippocampal–cortical coherence was higher during the rising phase in both NREM and WAKE, it is not likely that the relative lack of activity in the rising phase, which is not found broadly in WAKE, is behind it. Furthermore, units with a specific phase preference for the rising phase of ultraslow 5-HT oscillations represented a significant, if less numerous, group than units with a falling phase preference (*Figure 5—figure supplement 2*).

## Discussion

After simultaneous recordings of hippocampal 5-HT levels and LFP across behavioral states, we observed prominent ultraslow oscillations of 5-HT, which timed the occurrence of several electrophysiological read-outs. Specifically, we found that substates, similar to those described in NREM by previous studies (*Lecci et al., 2017*; *Osorio-Forero et al., 2021*), were closely linked to the phase of 5-HT ultraslow oscillations. Hippocampal ripples occurred preferentially during the falling phase of 5-HT oscillations, whereby hippocampal–cortical coherence was strongest and MAs and icEMG peaks were most prevalent during the rising phase. Importantly, these 5-HT-defined substates were observed to coordinate local and global brain activity not only within NREM, but also during WAKE states.

The prominence of internally driven ultraslow 5-HT oscillations during NREM explains why many studies focus on these rhythms during sleep. Potentially due to the requirement of WAKE to integrate external signals, ultraslow 5-HT oscillations appear less prominently in waking behavior. Nevertheless, our data shows that these ultraslow rhythms also play a role in WAKE, albeit to a lesser extent than in NREM. Indeed, studies have shown that the phase of ultraslow EEG oscillations during WAKE explains fluctuations in behavioral performance and arousal (*Fox et al., 2007*; *Monto et al., 2008*; *Sihn and Kim, 2022*). In our study, including both NREM and WAKE periods allowed us to additionally show that the temporal organization of activity relative to ultraslow 5-HT oscillations operates according to the same principles in both states, namely by segregating high arousal activity from low arousal internal processing. Rather than a special feature of NREM, the ultraslow 5-HT oscillation appears to be a more fundamental rhythm structuring brain activity. Along these lines, a recent study reported ultraslow oscillations in the firing of medial entorhinal cortex neurons which persisted throughout movement and immobility periods, substates of waking behavior (*Gonzalo Cogno et al., 2024*).

In order for the ultraslow oscillation phase to segregate brain activity, as we have observed, the hippocampal network must somehow be able to sense the direction of change of serotonin levels.

While single-cell mechanisms related to membrane potential dynamics are typically too fast to explain this calculation, theoretical work has suggested that feedback circuits can enable such temporal differentiation, also on the slower time scales we observe (*Tripp and Eliasmith, 2010*). Beyond the direction of change in serotonin levels, temporal differentiation could also enable the hippocampal network to discern the steeper rising slope versus the flatter descending slope that we observe in the ultraslow 5-HT oscillations (*Figure 1—figure supplement 1*), which may also be functionally relevant (*Cole and Voytek, 2017*). The distinction between the rising and falling phase of ultraslow oscillations is furthermore clearly discernible at the level of unit responses, with many units showing preferences for either half of the ultraslow period (*Figure 5—figure supplement 2*). Another factor that could help distinguish the rising from the falling phase is the level of other neuromodulators, as it is likely the combination of many neuromodulators at any given time that defines a behavioral substate. Given the finding that ACh and Oxt exhibit ultraslow oscillations with a temporal shift (*Zhang et al., 2024*), one could posit that distinct combinations of different levels of neuromodulators could segregate the rising from the falling phase via differential effects of the combination of neuromodulators on the hippocampal network.

In our data, we observed that the rising phase of ultraslow 5-HT oscillations was linked to arousal-associated brain activity. Specifically, long-range coherence, which has been shown to correlate with arousal and behavioral performance (*Klaver et al., 2023*; *Parto-Dezfouli et al., 2023*), was seen to peak in the rising phase of ultraslow 5-HT oscillations across a broad range of frequencies. In this way, 5-HT can be observed to gate communication between the cortex and hippocampus, reducing such communication during the ripple-associated falling phase. This gating could serve to reduce sensory input during 'internal' hippocampal memory processing, effectively decreasing potential interference that would disrupt memory consolidation (*Logothetis et al., 2012*; *Yang et al., 2019*). While the mechanism of 5-HT's effect on long-range neural synchrony is not yet clear, it has been shown that 5-HT can alter sensory gating dependent on communication between the thalamus and hippocampus (*Lee et al., 2020*). Furthermore, a recent study showed that type 2 dentate spikes (DS2s) in the hippocampus constitute substates of high arousal within immobility periods (*Farrell et al., 2024*). During DS2s, greater brain-wide activation was observed than during ripples, which mirrors the higher inter-areal coherence and likely higher arousal we observed during the rising phase of ultraslow 5-HT oscillations compared to the falling phase, when ripples preferentially occurred. The role of arousal-associated axo-axonic cells in producing DS2s suggests that these inhibitory cells may be a good target to further examine changes in arousal relative to ultraslow 5-HT oscillations. Indeed, we observed a subset of putative interneurons firing phase-locked to the rising phase (S6), although future work would be required to establish their identity.

MAs constitute periods of heightened arousability within NREM, which have been hypothesized to maintain a link between the sleeper and the outside world (*Halász et al., 2004*). That 5-HT signaling can affect MA occurrence has been shown in a psilocin study (*Thomas et al., 2022*). Furthermore, given the known association of MAs with increased hippocampal–cortical synchrony (*dos Santos Lima et al., 2019*), it is not surprising that they, as well as their associated icEMG peaks, like coherence, show a preference for the rising phase of ultraslow 5-HT oscillations. In fact, arousal from NREM sleep was shown to be more likely during NREM microstates with higher inter-cortical coherence (*Bastuji et al., 2021*). During WAKE, the icEMG signal was locked in the same way to ultraslow 5-HT oscillations, with peaks occurring during the rising phase. This link between serotonin and a correlate of movement is not surprising, as some studies have shown serotonergic control of movement in the hippocampus. Specifically, local infusion of 5-HT into the hippocampus was seen to induce locomotion, and serotonergic fibers in CA1 have been shown to activate upon movement initiation (*Takahashi et al., 2000*; *Luchetti et al., 2020*). Given that the phase of ultraslow serotonin oscillations times icEMG peaks, further investigating the reported link between 5-HT and repetitive movements in the context of 5-HT oscillations could shed more light on the question (*Alvarez et al., 2022*; *Jacobs and Fornal, 1991*).

According to studies performed to date, increasing 5-HT levels reduces ripple incidence both in vitro (*ul Haq et al., 2016*) and in vivo (*Wang et al., 2015*; *Shiozaki et al., 2023*). Based on these findings, one would expect a negative linear relationship in which ripples occur at the trough of 5-HT fluctuations, as was reported in the case of acetylcholine (*Yang et al., 2019*). In our study, the preference of ripples for the falling phase of ultraslow 5-HT oscillations shows that the dynamics of 5-HT changes are more determining for ripple occurrence than the absolute 5-HT level, at this time scale. However,

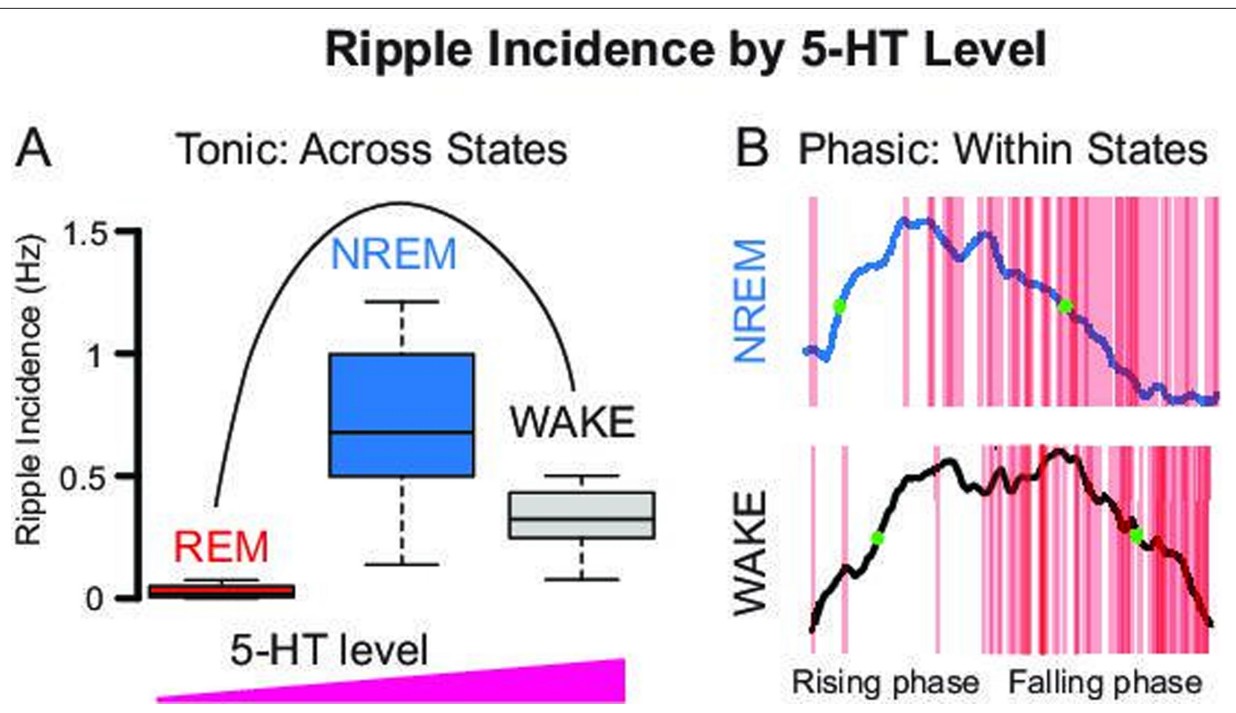

**Figure 6.** Relationship between ripple incidence and 5-HT levels depends on time scale. (**A**) Ripple incidence by behavioral state shows an inverted-U dose response relationship, with a peak at intermediate 5-HT levels (see *Figure 1F*). (**B**). Within states, ripple incidence depends on the phase of the ultraslow 5-HT oscillation. At the same absolute 5-HT level (e.g. green dots), therefore, different ripple incidences are observed.

The online version of this article includes the following figure supplement(s) for figure 6:

**Figure supplement 1.** Serotonin modulates ripple incidence in vitro.

when looking at a longer time scale, namely across states, the relationship between 5-HT level and ripple incidence shows an inverted-U shape with ripples occurring preferentially at the intermediate 5-HT levels observed in NREM (*Figure 6*). This inverted-U dose–response relationship was also found in preliminary hippocampal slice experiments, where the wash-in of low concentration 5-HT increased, and higher concentration 5-HT decreased, ripple incidence (*Figure 6—figure supplement 1*).

Existing support for the importance of 5-HT release dynamics in consequent brain activity and behavioral outcomes comes from a study showing different behavioral consequences of burst versus tonic 5-HT release (*Oikonomou et al., 2019*). Given their own findings that burst, but not tonic DRN stimulation induced waking, as well as studies showing that burst-firing of DRN neurons is associated with salient events (*Cohen et al., 2015*; *Paquelet et al., 2022*), Oikonomou et al. posited that 5-HT released in bursts is arousing, whereas tonic release controls slow behavioral state changes, such as increasing sleep drive during wake behavior. Along these lines, the regular increased burst firing observed in a subset of DRN neurons at ultraslow frequencies in vitro (*Mlinar et al., 2016*), likely corresponding to the rising phase of ultraslow 5-HT oscillations in our data, could be seen to signal the regular arousing signals which we observe in our ultraslow 5-HT oscillation-defined substates. Ambient 5-HT levels arising from slower changes in tonic state-dependent firing, on the other hand, could dictate the incidence range in which ripples can occur on a slower time scale, as in our preliminary slice experiments involving the slow wash-in of 5-HT.

A potential mechanism for how different 5-HT release dynamics could differentially affect the hippocampal network at the synaptic level comes from a study on extrasynaptic 5-HT release (*Trueta and De-Miguel, 2012*). In this study, high-frequency (10–20 Hz), but not low-frequency (1 Hz) stimulation was shown to induce extrasynaptic release of 5-HT in the leech Retzius neuron. The effect of such extrasynaptic release would be the targeting of receptors and/or neurons not affected by exclusively synaptic release, thus changing the overall network response to 5-HT. Further support for this idea comes from a recent study showing that during tonic and low-frequency phasic activity of serotonergic

neurons, synaptic transmission predominates, while during high-frequency and synchronized phasic activity, volume transmission occurs (*Zhang et al., 2025*).

While ripple incidence was biased to the falling phase of ultraslow 5-HT oscillations, higher power ripples were found to cluster around the peak. It follows that, during NREM, the peak of 5-HT oscillations could define a heightened period of ripple propagation to the cortex, which has been shown to be greater in higher power ripples throughout the brain (*Nitzan et al., 1947*), as well as within the hippocampus (*De Filippo and Schmitz, 2024*). In support of this claim, we found cortical power in high frequencies (100–150 Hz) to be greatest around the peak of hippocampal 5-HT oscillations (*Figure 5—figure supplement 1D*). As hippocampal–cortical interaction during ripples is thought to be a key factor in the consolidation of memory during NREM, the peak of ultraslow 5-HT oscillations could be seen to time memory consolidation itself (*Rothschild et al., 2017*; *Maingret et al., 2016*). Further studies are necessary to clarify the relationship between ultraslow 5-HT oscillations, ripple propagation, and memory processes.

The two previously mentioned in vivo studies showed reduced ripple incidence after *systemically* increasing 5-HT levels, either through intraperitoneal injections of an SSRI or global activation of median raphe nucleus neurons (*Wang et al., 2015*; *Shiozaki et al., 2023*). Given the regional specificity of the 5-HT system, such systemic activation has the potential to introduce effects both non-specific to the region of interest and potentially non-physiological. Systemic administration of a 5-HT$_4$ receptor agonist, for example, was shown to inhibit locomotion in an open field test, whereas local manipulation of CA1 terminals did not (*Teixeira et al., 2018*). Furthermore, different 5-HT fibers in CA1 were shown to be active during movement initiation and reward (*Luchetti et al., 2020*), indicating that even activating 5-HT fibers within one region at the same time has the potential to activate systems which are not naturally active together. Finally, the mode of systemic release has been shown to make a difference in the resulting behavioral outcome and neural response. In one study, phasic and chronic stimulation of the DRN were shown to inhibit and promote locomotion, respectively (*Correia et al., 2017*). In another study, DRN neurons were shown to have both immediate, i.e. phasic, responses to reward and punishment, but also adjust their tonic firing on the time scale of minutes (*Cohen et al., 2015*).

The conclusion of studies showing reduced ripples with increased 5-HT can further be understood in terms of physiological dose. Inhibitory 5-HT$_{1a}$ receptors have been shown to be expressed extra-synaptically in CA1 pyramids (*Riad et al., 2000*). Therefore, one could imagine that after in vitro bath application of high concentration 5-HT, or excess stimulation of 5-HT terminals, leading to extrasynaptic release and/or volume transmission of 5-HT, these receptors could be selectively engaged to silence pyramids and shut down the network. In fact, an inverted-U dose response curve with suppression at higher 5-HT levels was found both between 5-HT levels and ripples, in our preliminary in vitro wash-in experiments, as well as in a computational study of the effect of 5-HT on spatial working memory in the medial prefrontal cortex (*Cano-Colino et al., 2014*).

In addition to the difficulties involved with typical causal interventions already mentioned, the fact that the levels of different neuromodulators are inter-related and affected by ongoing brain activity makes it very hard to pinpoint ultraslow oscillations of one specific neuromodulator as controlling specific activity patterns, such as ripple timing. While a recent paper purported to show a causative effect of norepinephrine levels on ultraslow oscillations of sigma band power, the fact that optogenetic inhibition of locus coeruleus (LC) cells, but also excitation, only caused a minor reduction of the ultraslow sigma power oscillation suggests that other factors also contribute (*Osorio-Forero et al., 2021*). Generally, it is thought that many neuromodulators together determine brain states in a combinatorial manner, and it is probable that the 5-HT oscillations we measure, like the similar oscillations in NE, are one factor among many.

Nevertheless, given the known effects of 5-HT on neurons, it is not unlikely that the 5-HT fluctuations we describe have some impact on the timing of ripples, MAs, hippocampal–cortical coherence, or EMG signals that correlate with either the rising or descending phase. In fact, causal effects of 5-HT on ripple incidence (*Wang et al., 2015*; *ul Haq et al., 2016*; *Shiozaki et al., 2023*), MA frequency (*Thomas et al., 2022*), sensory gating (*Lee et al., 2020*), which is subserved by inter-areal coherence (*Fisher, 2020*), and movement (*Takahashi et al., 2000*; *Alvarez et al., 2022*; *Jacobs and Fornal, 1991*; *Luchetti et al., 2020*) have all been shown. Our added findings that serotonin affects ripple incidence in hippocampal slices in a dose-dependent manner (*Figure 6—figure supplement*

*1*) further suggest that the relationship between ultraslow 5-HT oscillations and ripples we report may indeed result, at least in part, from a direct effect of serotonin on the hippocampal network.

Whether these 'causal' relationships between 5-HT and the different activity measures we describe can be used to support a causal link between ultraslow 5-HT oscillations and the correlated activity we report remains an open question. To that point, some studies have described changes in ultraslow oscillations due to manipulation of serotonin signaling. Specifically, reduction of 5-HT$_{1a}$ receptors in the dentate gyrus was recently shown to reduce the power of ultraslow oscillations of calcium activity in the same region (*Turi et al., 2024*). Furthermore, psilocin, which largely acts on the 5-HT$_{2a}$ receptor, decreased NREM episode length from around 100 s to around 60 s and increased the frequency of brief awakenings (*Thomas et al., 2022*). While ultraslow oscillations were not explicitly measured in this study, the change in the rhythmic pattern of NREM sleep episodes and brief awakenings, or MAs, suggests an effect of psilocin on ultraslow oscillations during NREM. Although these studies do not necessarily point to an exclusive role for 5-HT in controlling ultraslow oscillations of different brain activity patterns, they show that changes in 5-HT can contribute to changes in brain activity at ultraslow frequencies.

These methodological considerations highlight the benefit of the correlative approach adopted here, measuring local 5-HT levels and brain activity simultaneously. While causal relationships cannot be determined from our in vivo strategy, the relationship observed between 5-HT levels and ripples can be used to inform future causal studies in a data-driven way. For example, the findings highlight the importance of having a detailed look at the relationship between different 5-HT release dynamics and hippocampal cell and network responses. Furthermore, modulating the frequency or strength of the ultraslow 5-HT oscillations, as done in *Osorio-Forero et al., 2021* for spindle oscillations and noradrenaline, could provide insight into how 5-HT tone and phasic release modulate ripples in a realistic setting.

## Materials and methods

### Animals

All experiments were performed in accordance with regulations of the Landesamt für Gesundheit und Soziales (Berlin [T0100/03], Berlin [G0189/17]) and European legislation (European Directive 2010/63/EU).

All experimental procedures were performed following the Guide for Animal Care and Use of Laboratory Animals. Male C57BL/6J mice (Jackson Laboratory) between 2 and 6 months of age were used for experiments. The mice were housed in groups of two to five animals prior to surgery, and singly after surgery, in a reverse 12/12 hr light–dark cycle (lights on 10 p.m. to 10 a.m.) with ad libitum access to food and water.

### Surgery

For viral injection, as well as for the implantation of optic fiber cannulae and silicon probes, mice were anesthetized with isoflurane (4%) and placed in a stereotactic frame (Kopf Instruments). Body temperature was maintained at 38°C by a heating pad (Harvard Instruments). The isoflurane level was slowly reduced to 1–2% to maintain anesthesia throughout the surgery. Mice were injected with ketoprofen (10 mg/kg, s.c.). Hair was removed with a depilatory cream, the scalp was cleaned with ethanol and iodine solutions, and the skull was exposed.

### Viral injection and optic fiber implantation

A craniotomy was performed by drilling a small hole above the right dorsal CA1 (AP –2.3 mm, ML –2.00 mm). A glass injection micropipette with a 100-µm tip was pulled, filled with mineral oil, and connected to a Hamilton syringe attached to a microsyringe pump (KD Scientific, Legato 111). 250 nl of AAV9-hSyn-5HT3.0 (WZ Biosciences) was injected at a rate of 100 nl/min and a depth of 1.3 mm below the dura. After injection, the pipette was left in place for 5 min before slowly bringing it up out of the brain over the course of 20–30 min. Saline was administered to the craniotomy site to keep the tissue moist throughout the procedure.

An optic fiber (Thorlabs CFML15L05) was then implanted above the injection site. Only fibers with >80% transmission efficacy were used. The optic fiber was secured with C&B Metabond (Parkell).

Dental cement was applied to exposed areas of the skull. Mice were kept in their home cages for 3 weeks to allow recovery from surgery and expression of the virus.

## Silicon probe implantation and electrophysiological recordings

Three weeks after viral injection and optic fiber implantation, a second surgery was performed to implant a silicon probe (NeuroNexus, 64-channel, edge, 1 or 4 shanks), mounted onto a microdrive (NeuroNexus, dDrive) into the left dorsal CA1. To this end, anesthesia was induced, as previously described, and a mouse cap with copper mesh (3DNeuro) was cemented to the skull. A second craniotomy was then performed over the right dorsal CA1 (AP –2.3 mm, ML +2.00 mm). The probe was slowly lowered and secured with C&B Metabond at a depth of 0.8 mm below the dura. The exposed brain was covered with a mixture of heated bone wax and mineral oil. A grounding screw was placed over the cerebellum and soldered to the ground electrode on the probe and the mouse cap. The mouse was allowed to recover for 5 days, at which point the probe was further lowered until the prominent spikes and sharp wave ripples characteristic of the CA1 pyramidal layer were observed. Data was recorded with an Open Ephys system at 20 kHz.

## Fiber photometry

Fiber photometry was performed as described in *Krok et al., 2023*. Briefly, a fiber-coupled 470 nm LED (Thorlabs) was used to send excitation light in continuous wave mode through a fiber optic patch cord (Doric) to the mouse's optic fiber implant via a fluorescence mini-cube (Doric). Emitted light traveled back through the same optic fiber patch cord to the mini-cube and was collected by a photoreceiver (Newport, DC mode). Signal collected by the photoreceiver was digitized at 5 kHz with a National Instruments acquisition board (NI BNC-2090A) and analyzed using Wavesurfer (Janelia).

Preprocessing of fiber photometry data was performed as described by Thomas Akam (Github, Thomas Akam, photometry_preprocessing: https://github.com/ThomasAkam/photometry_preprocessing/tree/master, *Akam and Burgeno, 2023*). Namely, raw fiber photometry data was first downsampled to 1250 Hz for comparison with electrophysiology data and low pass filtered at 20 Hz using MATLAB's (R2021a) lowpass function. Next, the slow drift in signal due to photobleaching was corrected by fitting a second-order exponential function. Finally, in order to compare photometry data across sessions and mice, the signals were z-scored.

## Fluoxetine injections

In order to show that the GRAB5-HT3.0 sensor used is responsive to changes in local 5-HT levels, 10 mg/kg fluoxetine-hydrochloride (Sigma-Aldrich) dissolved in saline or saline only (control) was administered intraperitoneally after a 20-min baseline.

## Dual fiber photometry and silicon probe recordings

Contralateral simultaneous recordings were chosen over ipsilateral due to the size of the optic fibers and fragility of the silicon probes, which prevented their implantation in close proximity. In addition to its adoption in a recent dual-recording study (*Zhang et al., 2024*), this contralateral recording scheme can be justified due to the simultaneous occurrence of ripples, a major electrophysiological read-out in the current study, as well as dentate spikes across hemispheres (*Chrobak and Buzsáki, 1996*; *Buzsáki et al., 2003*, *Farrell et al., 2024*), bilateral synchrony of ultraslow EEG oscillations (*Liu et al., 2010*), as well as the bilateral symmetry of the 5-HT system (*Ren et al., 2018*).

Fiber photometry and electrophysiological data were simultaneously recorded from double-implanted mice in their home cages for 2–3 hr sessions, containing both wake and sleep periods. Synchronization of photometry and electrophysiological data was performed by triggering recording onset with an Arduino pulse.

## CNN for ripple detection

The custom CNN model used for ripple detection was inspired by the approach of *Navas-Olive et al., 2022* and can be found on GitHub (https://github.com/clairecooper2193/ripNet) (*Cooper, 2024*). In their work, ripple detection was reframed from 1D thresholding of spectral features over time to an image recognition problem, where the image consists of segments of LFP data from multiple channels containing ripples. Detection thus takes into account the classic laminar pattern of ripples, which

is useful for distinguishing them from other fast oscillations or noise. Furthermore, this approach is unbiased in the sense that it does not rely upon a strict limitation of ripple features and, furthermore, has been shown to perform more consistently with data from different experimental sessions than standard filter-based detection (*Navas-Olive et al., 2024*).

## Data preparation

In constructing our model, which is structurally simpler than that proposed in *Navas-Olive et al., 2022*. Inputs to the network were prepared as follows: four neighboring channels in CA1 showing ripples were chosen. With 50 μm spacing between electrode sites, four channels showing ripple oscillations displayed characteristic amplitude changes that were key to distinguishing them from movement artifacts, fast gamma oscillations, or other false positives often detected as ripples by traditional ripple detection algorithms. Additionally, four channels were chosen from the neocortex to increase the network's ability to rule out movement artifacts or other noise which propagates uniformly across channels. Data from the eight channels was then *z*-scored and segmented into 400 ms chunks, and these 8 channel × 400 ms chunks were fed in as 2D inputs to the network (*Figure 2A*).

## Ripple annotation and training data

To prepare a training set, ripple start and stop times were labeled for 5000 ripples occurring in data from three different mice. A key criterion for distinguishing ripples from non-ripples was signal modulation across hippocampal and cortical channels. Fast and transient oscillations in which the amplitude varied according to location relative to the center of the CA1 pyramidal layer were considered ripples, whereas oscillations in which the amplitude appeared constant across hippocampal or hippocampal and cortical channels were excluded. 400 ms data segments centered around the labeled ripples were extracted for the eight input channels (four hippocampal and four cortical). A label trace (1 × 400 ms) was generated for each segment in which time outside of ripples was taken to be 0, and time during ripples, 1. For training, 5000 negative examples, that is data segments including no ripples, were also included. The timing of these segments was chosen at random such that there was no overlap between them and the ripple-containing segments and was taken from the same eight channels and three mice used for ripple-containing segments. Label traces were generated for each negative example consisting of a 1 × 400 ms trace of zeroes. Training data was split into a training set (80%) and a testing set (20%).

## CNN architecture

A custom CNN was built using Python 3.9.12 and the following key python packages:

    tensorflow 2.9.1
    keras 2.9.0
    numpy 1.23.0
    scipy 1.8.1

The model was built within a custom model class called RipNet. RipNet consisted of four convolution blocks, with each block consisting of a 2D convolutional layer (Conv2D), followed by a Rectified Linear Unit (ReLU) activation layer and a Batch Normalization layer. Stride length was (2,2) for all blocks, and kernel size was (1,1) for the first block, and (3,3) for the subsequent three blocks. The convolution blocks were followed by a Dropout layer (0.25), a Dense layer, a Batch Normalization layer, and a second Dropout layer (0.4). A sigmoid activation function on the final Dense layer provided the prediction trace, which gave the likelihood of ripple occurring as a number between 0 and 1, for the input trace provided. Ripples were detected when the prediction trace exceeded an empirically determined threshold of 0.2. Ripple peak times were determined by taking the peak of the envelope of the ripple-band (120–250 Hz) bandpass-filtered signal.

Network architectures, that is the number of convolutional blocks, and hyperparameters, including the optimizer, learning rate, and regularization, were tuned during training. After training, the best-performing model consisted of four convolutional blocks, an Adam optimizer with a learning rate of $1e-4$ and a decay rate of $1e-4$, as well as L2 regularization (0.001) to prevent overfitting. Mean-squared error was used as a loss function to compare the predicted trace to the ground truth trace.

## Model performance

Model performance was evaluated based on ripples detected in 2 hr of data across two mice, which was not included in the training dataset. In this testing set, ripples were annotated manually and compared to model predictions. True positives (TPs) occurred when manually labeled ripples were also predicted by the model. False negatives (FNs) were marked where ripples were annotated, but not predicted by the model. False positives (FPs) encompassed ripples predicted by the model and not labeled manually, and upon second inspection, not considered ripples. Based on these metrics, the F1 value, which represents the harmonic mean of precision (TP/TP + FP) and recall (TP/TP + FN), was calculated as a measure of model performance. The F1 value was found to be 81.5% for the testing data set. Of note, this F1 value is higher than that reported for both the standard Butterworth filter method at optimized performance and the aforementioned previously published CNN (*Navas-Olive et al., 2022*), with F1 values of 68% and 65%, respectively. To ensure that our model did not systematically exclude ripples with certain features in a way that would influence their overall relationship to ultraslow 5-HT oscillations, we took a closer look at the ripples missed by the model. Specifically, we compared ripple power in these missed ripples to power in correctly identified ripples (*Figure 2—figure supplement 1A, B*). Ultimately, we showed that, despite the model performing slightly better during the falling phase, as indicated by higher recall, inclusion of these missed ripples did not affect the fundamental relationship between ripples and ultraslow 5-HT oscillations (*Figure 2—figure supplement 1C*).

## State scoring

Behavioral states were designated as WAKE, NREM, REM, or MA according to the output of the SleepScoreMaster function from the Buzsaki lab code repository (https://github.com/buzsakilab/buzcode; copy archived at *Buzsaki Lab, 2023*). The methodology for SleepScoreMaster's sleep score detection is described in *Watson et al., 2016*. Briefly, the LFP power at low frequencies (<25 Hz) was first used to distinguish NREM from 'other' states. Next, the 'other' states were assigned labels according to the narrow theta-band ratio (5–10 Hz/2–6 Hz) and the icEMG, with high theta-band ratio and low EMG corresponding to REM states, and remaining states being classified as WAKE (>40 s) or MA (<40 s). Detected states were then reviewed manually by the experimenter. The icEMG was calculated as part of the SleepScoreMaster function as described in *Watson et al., 2016*. Briefly, the correlations between the signals from spatially distal pairs of channels, bandpass filtered from 300 to 600 Hz, were computed in 0.5 s bins. The average of these correlations was then taken as the icEMG. Notably, the icEMG, as calculated here, and traditional EMG signals show a high correlation (*Schomburg et al., 2014*), and discrepancies between the two measures were shown to reflect, for example, cases where the animal's face moved with no overall head displacement, such as occurs during chewing (*Watson et al., 2016*).

## LFP analysis

LFP analysis was performed using custom MATLAB code. Time–frequency power spectra were generated using the Stockwell Transform (*Stockwell et al., 1996*) with the function st (*Sundar, 2015*). Time–frequency power spectra were normalized to 0–1 for visualization. Phase angles were calculated using the Hilbert transform of the bandpass-filtered signal (0.01–0.06 Hz). Magnitude-squared wavelet coherence was calculated using MATLAB's wcoherence function.

## Histology

Transcardiac perfusion was performed after deep anesthesia with isoflurane with 4% paraformaldehyde (PFA) in 0.1 M sodium phosphate buffer. Brains were kept in PFA overnight, then sliced into 100 μm coronal sections with a vibratome (Leica). Slices were mounted using Fluoroshield with DAPI (Sigma), and endogenous fluorescence from the GRAB5-HT3.0 sensor was imaged with an Olympus VS120 slide scanning microscope. From these images, the expression of the GRAB5-HT3.0 sensor in the dorsal CA1 was verified.

## Statistics

All plots with error bars or bounded lines reflect the mean across sessions ± standard error of the mean. Statistics were performed in RStudio, using R version 4.3.2. As the data for all experiments is hierarchical, it is necessary to account for inter-mouse and inter-session variation (*Yu et al., 2022*).

To this end, we fit GLMMs, including a fixed effect term of state (NREM and WAKE) and random effects terms for mouse and session ID. Unless otherwise specified, p values were obtained with *emmeans*, which was used for post hoc testing of fitted models (*Lenth, 2024*) as well as a Bonferroni adjustment for multiple comparisons. For *Figure 1F*, the *blme* package was used with an identity link function (*Chung et al., 2013*). For *Figures 3F2, F4, and 4C*, a logit link function was used with the lme4 package (*Bates et al., 2015*). Significant phase preference for the reference state (NREM) was inferred from the significance of the intercept. To determine whether the non-reference state (WAKE) also exhibited a significant bias, we calculated its log-odds by adding the intercept and the state coefficient. The corresponding *z*-score was computed, and the p-value was then obtained from a two-tailed normal distribution. As MAs occur only during NREM, state was no longer a fixed effect, and significance was inferred from the significance of the model's intercept. For Figure S2B, glmer() from *lme4* was used with an identity function (*Bates et al., 2015*). For *Figures 1H and 3H*, S1 and S5, a log link function was used with the *glmmTMB* package (*Brooks et al., 2017*). For *Figures 4G and 5D*, a beta distribution link function was used with the *glmmTMB* package. For Figure S2D, the *lme4* package was used (*Bates et al., 2015*). Ratios were log-transformed such that the significance of the intercept reflected the difference of non-transformed ratios from 1. For *Figure 1I*, a Wilcoxon ranked-sum test was used on the mean 5-HT levels in fluoxetine and saline-injected mice in the 20-min period starting 20 min after the injection.

## Acknowledgements

We would like to thank the Buzsàki lab for providing the virus expressing the GRAB5-HT3.0 sensor used in this study, and Dr. Yiyao Zhang for offering some initial training and input on fiber photometry. Additionally, we are grateful to the members of the Tritsch lab for their insightful discussions, sharing lab equipment, and assistance with fiber photometry, especially James Taniguchi and Tony Garcia. Funding This study was supported by the German Research Foundation (Deutsche Forschungsgemeinschaft (DFG), project 184695641 – SFB 958, project 327654276 – SFB 1315, project 415914819 – FOR 3004, project 431572356, and under Germany's Excellence Strategy – Exc-2049-390688087), by the Federal Ministry of Education and Research (BMBF, SmartAge-project 01GQ1420B) and by the European Research Council (ERC) under the European Union's Horizon 2020 research and innovation program (grant agreement No. 810580).

## Additional information

### Funding

| Funder | Grant reference number | Author |
|---|---|---|
| Deutsche Forschungsgemeinschaft | 184695641 | Dietmar Schmitz |
| Deutsche Forschungsgemeinschaft | 327654276 | Dietmar Schmitz |
| Deutsche Forschungsgemeinschaft | 431572356 | Dietmar Schmitz |
| Deutsche Forschungsgemeinschaft | Exc-2049-390688087 | Dietmar Schmitz |
| Bundesministerium für Bildung und Forschung | 01GQ1420B | Dietmar Schmitz |
| European Research Council | 10.3030/810580 | Dietmar Schmitz |

The funders had no role in study design, data collection, and interpretation, or the decision to submit the work for publication.

## Author contributions

Claire Cooper, Conceptualization, Data curation, Formal analysis, Investigation, Writing – original draft, Writing – review and editing; Daniel Parthier, Resources, Formal analysis, Writing – review and editing; Jeremie Sibille, John J Tukker, Nicolas Tritsch, Resources, Writing – review and editing; Dietmar Schmitz, Conceptualization, Supervision, Funding acquisition, Investigation, Project administration, Writing – review and editing

## Author ORCIDs

Claire Cooper ⓘ https://orcid.org/0000-0002-1279-4800
Daniel Parthier ⓘ http://orcid.org/0000-0001-8775-024X
Jeremie Sibille ⓘ http://orcid.org/0000-0001-6895-7405
John J Tukker ⓘ https://orcid.org/0000-0002-4394-199X
Nicolas Tritsch ⓘ https://orcid.org/0000-0003-3181-7681
Dietmar Schmitz ⓘ https://orcid.org/0000-0003-2741-5241

## Ethics

All animal experiments were carried out according to the guidelines stated in Directive 2010/63/EU of the European Parliament on the protection of animals used for scientific purposes and were approved by the animal welfare committee of Charité – Universitätsmedizin Berlin and the Landesamt für Gesundheit und Soziales Berlin (permit G 0298/18).

Reviewer #1 (Public review): https://doi.org/10.7554/eLife.101105.3.sa1
Reviewer #2 (Public review): https://doi.org/10.7554/eLife.101105.3.sa2
Reviewer #3 (Public review): https://doi.org/10.7554/eLife.101105.3.sa3
Author response https://doi.org/10.7554/eLife.101105.3.sa4

---

# Additional files

## Supplementary files

MDAR checklist

## Data availability

Data required to reproduce findings can be found on FigShare: https://doi.org/10.6084/m9.figshare.25794694.v3.

The following dataset was generated:

| Author(s) | Year | Dataset title | Dataset URL | Database and Identifier |
|---|---|---|---|---|
| Cooper C | 2024 | Fiber Photometry | https://doi.org/10.6084/m9.figshare.25794694.v3 | figshare, 10.6084/m9.figshare.25794694.v3 |

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
