## [Editor Report · eLife Assessment]

This **important** study demonstrates that slow fluctuations in serotonin release during wakefulness and non-REM sleep correspond to periods of heightened arousal or enhanced offline information processing. The evidence supporting this claim is **convincing**, and the methodology is robust and broadly applicable, likely to benefit many researchers in the field. This work will be of significant interest to neuroscientists studying sleep, memory, and neuromodulation.

---

## [Referee Report · Reviewer #1 (Public review)]

Summary:

In this work, authors recorded the dynamics of the 5-HT with fiber photometry from CA1 in one hemisphere and LFP from CA1 in the other hemisphere. They have observed an ultra-slow oscillation in the 5-HT signal both during wakefulness and NREM sleep. The authors have studied different phases of the ultra-slow oscillation to examine the potential difference in the occurrence of some behavioral state-related physiological phenomena (hippocampal ripples, EMG, and inter-area coherence).

Strengths:

The relation between the falling/rising phase of the ultra-slow oscillation and the ripples is sufficiently shown. There are some minor concerns about the observed relations that should be addressed with some further analysis.

Systematic observations have started to establish a strong relation between the dynamics of neural activity across the brain and measures of behavioral arousal. Such relations span a wide range of temporal scales that are heavily inter-related. Ultra-slow time scales are specifically understudied due to technical limitations and neuromodulatory systems are the strongest mechanistic candidates for controlling/modulating the neural dynamics at these time scales. The hypothesis of the relation between a specific time scale and one certain neuromodulator (5-HT in this manuscript) could have a significant impact on the understanding of the hierarchy in the temporal scales of neural activity.

Weaknesses:

weaknesses appropriately addressed by reviewers in the current version

---

## [Referee Report · Reviewer #2 (Public review)]

Summary:

In their study, Cooper et al. investigated the spontaneous fluctuations in extracellular 5-HT release in the CA1 region of the hippocampus using GRAB5-HT3.0. Their findings revealed the presence of ultra-low frequency (less than 0.05 Hz) oscillations in 5-HT levels during both NREM sleep and wakefulness. The phase of these 5-HT oscillations was found to be related to the timing of hippocampal ripples, microarousals, electromyogram (EMG) activity, and hippocampal-cortical coherence. In particular, ripples were observed to occur with greater frequency during the descending phase of 5-HT oscillations, and stronger ripples were noted to occur in proximity to the 5-HT peak during NREM. Microarousal and EMG peaks occurred with greater frequency during the ascending phase of 5-HT oscillations. Additionally, the strongest coherence between the hippocampus and cortex was observed during the ascending phase of 5-HT oscillations. These patterns were observed in both NREM sleep and the awake state, with a greater prevalence in NREM. The authors posit that 5-HT oscillations may temporally segregate internal processing (e.g., memory consolidation) and responsiveness to external stimuli in the brain.

Strengths:

The findings of this research are novel and intriguing. Slow brain oscillations lasting tens of seconds have been suggested to exist, but to my knowledge they have never been analyzed in such a clear way. Furthermore, although it is likely that ultra-slow neuromodulator oscillations exist, this is the first report of such oscillations, and the greatest strength of this study is that it has clarified this phenomenon both statistically and phenomenologically.

Weaknesses:

As with any paper, this one has some limitations. While there is no particular need to pursue them, I will describe ten of them below, including future directions:

Contralateral recordings: 5-HT levels and electrophysiological recordings were obtained from opposite hemispheres due to technical limitations. Ipsilateral simultaneous recordings may show more direct relationships.

Sample size: The number of mice used in the experiments is relatively small (n=6). Validation with a larger sample size would be desirable.

Lack of causality: The observed associations show correlations, not direct causal relationships, between 5-HT oscillations and neural activity patterns.

Limited behavioral states: The study focuses primarily on sleep and quiet wakefulness. Investigation of 5-HT oscillations during a wider range of behavioral states (e.g., exploratory behavior, learning tasks) may provide a more complete understanding.

Generalizability to other brain regions: The study focuses on the CA1 region of the hippocampus. It's unclear whether similar 5-HT oscillation patterns exist in other brain regions.

Long-term effects not assessed: Long-term effects of ultra-low 5-HT oscillations (e.g., on memory consolidation or learning) were not assessed.

Possible species differences: It's uncertain whether the findings in mice apply to other mammals, including humans.

Technical limitations: The temporal resolution and sensitivity of the GRAB5-HT3.0 sensor may not capture faster 5-HT dynamics.

Interactions with other neuromodulators: The study does not explore interactions with other neuromodulators (e.g., norepinephrine, acetylcholine) or their potential ultraslow oscillations.

Limited exploration of functional significance: While the study suggests a potential role for 5-HT oscillations in memory consolidation and arousal, direct tests of these functional implications are not included.

---

## [Referee Report · Reviewer #3 (Public review)]

Summary:

Activity of serotonin (5-HT) releasing neurons as well as 5-HT levels in brain structures targeted by serotoninergic axons are known to fluctuate substantially across the animal's sleep/wake cycle, with high 5-HT during wakefulness (WAKE), intermediate 5-HT levels during non-REM sleep (NREM) and very low 5-HT levels during REM sleep. Recent studies have shown that during NREM, activity of 5-HT neurons in raphe nuclei oscillates at very low frequencies (0.01 - 0.05 Hz) and this ultraslow oscillation is negatively coupled to broadband EEG power. However, how exactly this 5-HT oscillation affects neural activity in downstream structures is unclear.

The present study addresses this gap by replicating the observation of the ultraslow oscillation in the 5-HT system, and further observing that hippocampal sharp wave-ripples (SWRs), biomarkers of offline memory processing, occur preferentially in barrages on the falling phase of the 5-HT oscillation during both wakefulness and NREM sleep. In contrast, the study found that the raising phase of the 5-HT oscillation is associated with microarousals during NREM and increased muscular activity during WAKE. Finally, the raising 5-HT phase was also found to be associated with increased synchrony between the hippocampus and neocortex.

In vivo findings are further supported by an ex vivo demonstration of dose-dependent serotonergic SWR modulation, lends support to the potential causal relationship between 5-HT slow oscillation and hippocampal dynamics.

Overall, the study constitutes a valuable contribution to the field by reporting a close association between, on one hand, raising 5-HT and arousal and, on the other hand, falling 5-HT and offline memory processes.

Strengths:

The study makes a compelling use of the state-of-the art methodology to address its aims: the genetically encoded 5-HT sensor used in the study is ideal for capturing the ultraslow 5-HT dynamics and the novel detection method for SWRs outperforms current state-of-the-art algorithms and will be useful to many scientists in the field. Explicit validation of both of these methods is a particular strength of this study.

The analytical methods used in the article are appropriate and are convincingly applied, the use of a general linear mixed model for statistical analysis is a particularly welcome choice as it guards against pseudoreplication while preserving statistical power.

Pharmacological demonstration of serotonergic SWR modulation in brain slices adds further weight to the possible direct role of 5-HT in hippocampal dynamics in vivo.

Overall, the manuscript makes a strong case for distinct sub-states across WAKE and NREM, associated with different phases of the 5-HT oscillation.

Weaknesses:

All in vivo evidence presented in the study is correlational, although the ex vivo results do suggest a possibility of a causal relationship between 5HT levels and hippocampal dynamics in the intact brain.

---

## [Author Response]

The following is the authors’ response to the original reviews.

**Public Reviews:**

**Reviewer #1 (Public review):**
SummaryIn this work, the authors recorded the dynamics of the 5-HT with fiber photometry from CA1 in one hemisphere and LFP from CA1 in the other hemisphere. They observed an ultra-slow oscillation in the 5-HT signal during both wake fulness and NREM sleep. The authors have studied different phases of the ultra-slow oscillation to examine the potential difference in the occurrence of some behavioral state-related physiological phenomena hippocampal ripples, EMG, and inter-area coherence.StrengthsThe relation between the falling/rising phase of the ultra-slow oscillation and the ripples is sufficiently shown. There are some minor concerns about the observed relations that should be addressed with some further analysis.Systematic observations have started to establish a strong relation between the dynamics of neural activity across the brain and measures of behavioral arousal. Such relations span a wide range of temporal scales that are heavily inter-related. Ultra-slow time-scales are specifically under-studied due to technical limitations and neuromodulatory systems are the strongest mechanistic candidates for controlling/modulating the neural dynamics at these time-scales. The hypothesis of the relation between a specific time-scale and one certain neuromodulator (5-HT in this manuscript) could have a significant impact on the understanding of the hierarchy in the temporal scales of neural activity.Weaknesses:One major caveat of the study is that different neuromodulators are strongly correlated across all time scales and related to this, the authors need to discuss this point further and provide more evidence from the literature (if any) that suggests similar ultra-slow oscillations are weaker or lack from similar signals recorded for other neuromodulators such as Ach and NA.

The reviewer is correct to point out that the levels of different neuromodulators are often correlated. For example, most monoaminergic neurons, including serotonergic neurons of the raphe nuclei, show similar firing rates across behavioral states, firing most during wake behavior, less during NREM, and ceasing firing during ‘paradoxical sleep’ or REM (Eban-Rothschild et al 2018). Notably, other neuromodulators, such as acetylcholine (ACh), show the opposite pattern across states, with highest levels observed during REM, an intermediate level during wake behavior, and the lowest level during NREM (Vazquez et al. 2001). Despite these differences, ultraslow oscillations of both monoaminergic and non-monoaminergic neuromodulators, have been described, albeit only during NREM sleep (Zhang et al. 2024, Osorio-Ferero et al. 2021, Kjaerby et al. 2022). How ultraslow oscillations of different neuromodulators are related has been only recently explored (Zhang et al. 2024). In this study, dual recording of oxytocin (Oxt) and ACh with GRAB sensors showed that the levels of the two neuromodulators were indeed correlated at ultraslow frequencies with a 2 s temporal shift. Furthermore, this shift could be explained by a hippocampal-to-lateral septum intermediate pathway, in which the level of ACh causally impacts hippocampal activity, which then in turn controls Oxt levels. Given the known temporal relationship between ripples, ACh and Oxt, and now with our work, between ripples and 5-HT, one could infer the relative timing of ultraslow oscillations of ACh, Oxt and 5-HT. While dual recordings of norepinephrine (NE) and 5-HT have not been performed, a similar correlation with temporal shift could be hypothesized given the parallel relationships between NE and spindles (OsorioFerero et al. 2021), and 5-HT and ripples, with the known temporal delay between ripples and spindles (Staresina et al. 2023). The fact that the locus coerulus receives particularly dense projections from the dorsal raphe nucleus (Kim et al. 2004) further suggests that 5-HT ultraslow oscillations could drive NE oscillations. How exactly ultraslow oscillations of serotonin are related to ultraslow oscillations of different neuromodulators in different brain regions remains to be studied.

We have further addressed this question and how it relates to the issue of causality in the Discussion section of the manuscript (p. 13):

“In addition to the difficulties involved with typical causal interventions already mentioned, the fact that the levels of different neuromodulators are interrelated and affected by ongoing brain activity makes it very hard to pinpoint ultraslow oscillations of one specific neuromodulator as controlling specific activity patterns, such as ripple timing. While a recent paper purported to show a causative effect of norepinephrine levels on ultraslow oscillations of sigma band power, the fact that optogenetic inhibition of locus coerulus (LC) cells, but also excitation, only caused a minor reduction of the ultraslow sigma power oscillation suggests that other factors also contribute (Osorio-Forero et al., 2021). Generally, it is thought that many neuromodulators together determine brain states in a combinatorial manner, and it is probable that the 5-HT oscillations we measure, like the similar oscillations in NE, are one factor among many.

Nevertheless, given the known effects of 5-HT on neurons, it is not unlikely that the 5-HT fluctuations we describe have some impact on the timing of ripples, MAs, hippocampal-cortical coherence, or EMG signals that correlate with either the rising or descending phase. In fact, causal effects of 5-HT on ripple incidence (Wang et al. 2015, ul Haq et al. 2016 and Shiozaki et al. 2023), MA frequency (Thomas et al. 2022), sensory gating (Lee et al. 2020), which is subserved by inter-areal coherence (Fisher et al. 2020), and movement (Takahashi et al. 2000, Alvarez et al. 2022, Jacobs et al. 1991 and Luchetti et al. 2020) have all been shown. Our added findings that serotonin affects ripple incidence in hippocampal slices in a dose-dependent manner (Figure S1) further suggests that the relationship between ultraslow 5-HT oscillations and ripples we report may indeed result, at least in part, from a direct effect of serotonin on the hippocampal network.

Whether these ‘causal’ relationships between 5-HT and the different activity measures we describe can be used to support a causal link between ultraslow 5-HT oscillations and the correlated activity we report remains an open question. To that point, some studies have described changes in ultraslow oscillations due to manipulation of serotonin signaling. Specifically, reduction of 5-HT1a receptors in the dentate gyrus was recently shown to reduce the power of ultraslow oscillations of calcium activity in the same region (Turi et al. 2024). Furthermore, psilocin, which largely acts on the 5-HT2a receptor, decreased NREM episode length from around 100 s to around 60 s, and increased the frequency of brief awakenings (Thomas et al. 2022). While ultraslow oscillations were not explicitly measured in this study, the change in the rhythmic pattern of NREM sleep episodes and brief awakenings, or microarousals, suggests an effect of psilocin on ultraslow oscillations during NREM. Although these studies do not necessarily point to an exclusive role for 5-HT in controlling ultraslow oscillations of different brain activity patterns, they show that changes in 5-HT can contribute to changes in brain activity at ultraslow frequencies.”

A major question that has been left out from the study and discussion is how the same level of serotonin before and after the peak could be differentially related to the opposite observed phenomenon. What are the possible parallel mechanisms for distinguishing between the rising and falling phases? Any neurophysiological evidence for sensing the direction of change in serotonin concentration (or any other neuromodulator), and is there any physiological functionality for such mechanisms?

We have added a paragraph in the discussion to address how this differentiation of the 5-HT signal may be carried out (Discussion, paragraph #3, p. 10):

“In order for the ultraslow oscillation phase to segregate brain activity, as we have observed, the hippocampal network must somehow be able to sense the direction of change of serotonin levels. While single-cell mechanisms related to membrane potential dynamics are typically too fast to explain this calculation, a theoretical work has suggested that feedback circuits can enable such temporal differentiation, also on the slower timescales we observe (Tripp and Eliasmith, 2010). Beyond the direction of change in serotonin levels, temporal differentiation could also enable the hippocampal network to discern the steeper rising slope versus the flatter descending slope that we observe in the ultraslow 5-HT oscillations (Figure S2), which may also be functionally relevant (Cole and Voytek, 2017). The distinction between the rising and falling phase of ultraslow oscillations is furthermore clearly discernible at the level of unit responses, with many units showing preferences for either half of the ultraslow period (Figure S6). Another factor that could help distinguish the rising from the falling phase is the level of other neuromodulators, as it is likely the combination of many neuromodulators at any given time that defines a behavioral substate. Given the finding that ACh and Oxt exhibit ultraslow oscillations with a temporal shift (Zhang et al. 2024), one could posit that distinct combinations of different levels of neuromodulators could segregate the rising from the falling phase via differential effects of the combination of neuromodulators on the hippocampal network.”

Functionally, the ability to distinguish between the rising and falling phases of an oscillatory cycle is a form of phase coding. A well-known example of this can be seen in hippocampal place cells, which fire relative to the ongoing theta oscillations. The key advantage of phase coding is that it introduces an additional dimension, i.e. phase of firing, beyond the simple rate of neural firing. This allows for the multiplexing of information (Panzeri et al., 2010), enabling the brain to encode more complex patterns of activity. Moreover, phase coding is metabolically more efficient than traditional spike-rate coding (Fries et al., 2007).

**Reviewer #2 (Public review):**
Summary:In their study, Cooper et al. investigated the spontaneous fluctuations in extracellular 5-HT release in the CA1 region of the hippocampus using GRAB5-HT3.0. Their findings revealed the presence of ultralow frequency (less than 0.05 Hz) oscillations in 5-HT levels during both NREM sleep and wakefulness. The phase of these 5-HT oscillations was found to be related to the timing of hippocampal ripples, microarousals, electromyogram (EMG) activity, and hippocampal-cortical coherence. In particular, ripples were observed to occur with greater frequency during the descending phase of 5-HT oscillations, and stronger ripples were noted to occur in proximity to the 5-HT peak during NREM. Microarousal and EMG peaks occurred with greater frequency during the ascending phase of 5-HT oscillations. Additionally, the strongest coherence between the hippocampus and cortex was observed during the ascending phase of 5-HT oscillations. These patterns were observed in both NREM sleep and the awake state, with a greater prevalence in NREM. The authors posit that 5-HT oscillations may temporally segregate internal processing (e.g., memory consolidation) and responsiveness to external stimuli in the brain.Strengths:The findings of this research are novel and intriguing. Slow brain oscillations lasting tens of seconds have been suggested to exist, but to my knowledge they have never been analyzed in such a clear way. Furthermore, although it is likely that ultra-slow neuromodulator oscillations exist, this is the first report of such oscillations, and the greatest strength of this study is that it has clarified this phenomenon both statistically and phenomenologically.Weaknesses:As with any paper, this one has some limitations. While there is no particular need to pursue them, I will describe ten of them below, including future directions:(1) Contralateral recordings: 5-HT levels and electrophysiological recordings were obtained from opposite hemispheres due to technical limitations. Ipsilateral simultaneous recordings may show more direct relationships.

Although we argue that bilateral symmetry defines both the serotonin system and many hippocampal activity patterns (Methods: Dual fiber photometry and silicon probe recordings), we agree that ipsilateral recordings would be superior to describe the link between serotonin and electrophysiology in the hippocampus. In addition to noting that a recent study has adopted the same contralateral design (Zhang et al. 2024), we add a reference further supporting bilateral hippocampal synchrony, specifically of dentate spikes (Farrell et al. 2024). However, as functional lateralization has been recently proposed to underlie certain hippocampal functions in the rodent (Jordan 2020), future studies should ideally include both imaging and electrophysiology in a single hemisphere to guarantee local correlations rather than assuming inter-hemispheric synchrony. This could be accomplished using an integrated probe with attached optical fibers, as described in Markowitz et al. 2018, which is however technically more challenging and has, to our knowledge, not yet been implemented with fiber photometry recordings with GRAB sensors. Given the required separation of a few hundred micrometers between the probe shanks and the optical fiber cannula, it is important to consider whether the recordings are capturing the same neuronal populations. For example, there is a risk of recording electrical activity from dorsal hippocampal neurons while simultaneously measuring light signals from neurons in the intermediate hippocampus, which are functionally distinct populations (Fanselow and Dong 2009).

(2) Sample size: The number of mice used in the experiments is relatively small (n=6). Validation with a larger sample size would be desirable.

While larger sample sizes generally reduce the influence of random variability and minimize the impact of outliers on conclusions, our use of mixed-effects models mitigates these concerns by accounting for both inter-session and inter-mouse variability. With this approach, we explicitly model random effects, such as the variability between individual mice and sessions, alongside fixed effects (such as treatment), which ensures that our results are not driven by random fluctuations in a few individual mice or sessions. Furthermore, the inclusion of random intercepts and slopes in the models allows for the possibility that different animals and/or sessions have different baseline characteristics and respond to different degrees of magnitude to the treatment. In summary, while validating these findings with a larger sample size would certainly help detect more subtle effects, we are confident in the robustness of the conclusions presented.

(3) Lack of causality: The observed associations show correlations, not direct causal relationships, between 5-HT oscillations and neural activity patterns.

We agree that the data we present in this study is largely correlational and generally avoid claims of causality in the manuscript. In the Discussion section, we discuss barriers to interpreting typical causal interventions in vivo, such as optogenetic activation of raphe nuclei: “The two previously mentioned in vivo studies showing reduced ripple incidence…”(paragraph #10, pg. 12), as well as an added section on further causality considerations in the Discussion section of the manuscript (paragraph #12, pg. 13): “In addition to the difficulties involved with…”

Due to these barriers, as a first step, we wanted to describe how physiological changes in serotonin levels are correlated to changes in the hippocampal activity. Equipped with a deeper understanding of physiological serotonin dynamics, future studies could explore interventions that modulate serotonin in keeping with the natural range of serotonin fluctuations for a given state. On that point, another challenge which we have not mentioned in the manuscript is that modulating serotonin, or any neuromodulator’s levels, has the potential, depending on the degree of modulation, to transition the brain to an entirely different behavioral state. This then complicates interpretation, as one is not sure whether effects observed are due to the changes in the neuromodulator itself, or secondary to changes in state. At the same time, 5-HT activity drives networks which in return can change the release of other neurotransmitters, leading to indirect effects.

The results of our in vitro experiments suggest that a causal relationship between serotonin and ripples is possible (Figure S1). Though the hippocampal slice preparation is clearly an artificial model, it provides a controlled environment to isolate the effects of serotonin manipulation on the hippocampal formation, without the confounding influence of systemic 5-HT fluctuations in other brain regions. Notably, the dose-dependent effects of serotonin (5-HT) wash-in on ripple incidence observed in vitro closely mirror the inverted-U dose-response curve seen in our in vivo experiments across states, where small increases in serotonin lead to the highest ripple incidence, and both lower and higher levels correspond to reduced ripple activity. This parallel suggests that the gradual washing of serotonin in our in vitro system may mimic the tonic firing changes in serotonergic neurons that occur during state transitions in vivo. These findings underscore the importance of studying how different dynamics of serotonin modulation can differentially affect hippocampal network activity.

(4) Limited behavioral states: The study focuses primarily on sleep and quiet wakefulness. Investigation of 5-HT oscillations during a wider range of behavioral states (e.g., exploratory behavior, learning tasks) may provide a more complete understanding.

We agree that future studies should investigate a broader range of behavioral states. For this study, as we were focused on general sleep and wake patterns, our recordings were done in the home cage, and we limited ourselves to the basic behavioral states described in the paper. Future studies should be designed to investigate ultraslow 5-HT oscillations during different behaviors, such as continuous treadmill running. Specifically, a finer segregation of extended wake behaviors by level of arousal could greatly add to our understanding of the role of ultraslow serotonin oscillations.

(5) Generalizability to other brain regions: The study focuses on the CA1 region of the hippocampus. It's unclear whether similar 5-HT oscillation patterns exist in other brain regions.

Given the reported ultraslow oscillations of population activity in serotonergic neurons of the dorsal raphe nucleus (Kato et al. 2022) as well as the widespread projections of the serotonergic nuclei, we would expect a broad expression of ultraslow 5-HT oscillations throughout the brain. So far, ultraslow 5-HT oscillations have been described in the basal forebrain, as well as in the dentate gyrus, in addition to what we have shown in CA1 (Deng et al. 2024 and Turi et al. 2024). Furthermore, our results showing that hippocampal-cortical coherence changes according to the phase of hippocampal ultraslow 5-HT oscillations suggests that 5-HT can affect oscillatory activity either indirectly by modulating hippocampal cells projecting to the cortical network or directly by modulating the cortical postsynaptic targets. Given the heterogeneity in projection strength, as well as in pre- and postsynaptic serotonin receptor densities across brain regions (de Filippo & Schmitz, 2024), it would be interesting to see whether local ultraslow 5-HT oscillations are differentially modulated, e.g. in terms of oscillation power. Future studies investigating different brain regions via implantation of multiple optic fibers in different brain areas or using the mesoscopic imaging approach adopted in Deng et al. 2024, will be needed to examine the extent of spatial heterogeneity in this ultraslow oscillation.

(6) Long-term effects not assessed: Long-term effects of ultra-low 5-HT oscillations (e.g., on memory consolidation or learning) were not assessed.

While beyond the scope of our current study, we agree that an important next step would involve modulating the ultraslow serotonin oscillation after learning, and then examining potential effects on memory consolidation, presumably via changes in ripple dynamics, though many possibilities could explain potential effects. There, our results suggest it would be important to isolate effects due to the change in ultraslow oscillation features, rather than simply overall levels of 5-HT. To that end, it would be important to test different modulation dynamics, specifically modulating the oscillation strength, around a constant mean 5-HT level by carefully timed optogenetic stimulation/inhibition. Afterwards, showing a clear correlation between the strength of the 5-HT modulation and memory performance would be important to establishing the relationship, as done in Lecci et al 2017, where more prominent ultraslow oscillations of sigma power in the cortex during sleep, alongside a higher density of spindles, were correlated with better memory consolidation. Given the tight coupling of spindles and ripples during sleep, it is possible that a similar effect on memory consolidation would be observed following changes in ultraslow 5-HT oscillation power.

(7) Possible species differences: It's uncertain whether the findings in mice apply to other mammals, including humans.

We agree that the experiments should ultimately be replicated in humans. In the 2017 study by Lecci et al., the authors highlighted the shared functional requirements for sleep across species, despite apparent differences, such as variations in sleep volume. To explore these commonalities, the researchers conducted parallel experiments in both mice and humans, aiming to identify a universal organizing structure. They discovered that the ultraslow oscillation of sigma power serves this role, enabling both species to balance the competing demands of arousability and sleep imperviousness. Based on this finding, it is plausible that ultraslow oscillations of serotonin, which similarly modulate activity according to arousal levels, would serve a comparable function in humans.

(8) Technical limitations: The temporal resolution and sensitivity of the GRAB5-HT3.0 sensor may not capture faster 5-HT dynamics.

The kinetics of the GRAB5-HT3.0 sensor used in this study limit the range of serotonin dynamics we can observe. However, the ultraslow oscillations we measure reflect temporal changes on the scale of 20 s and greater, whereas the GRAB sensor we use has sub-second on kinetics and below 2 s off kinetics (Deng et al. 2024). Therefore, the sensor is capable of reporting much faster activity than the ultraslow oscillations we observe, indicating that the ultraslow 5-HT signal accurately reflects the dynamics on this time scale. Furthermore, the presence of ultraslow oscillations in spiking activity—observed in the hippocampal formation (Gonzalo Cogno et al., 2024; Aghajan et al., 2023; Penttonen et al., 1999) and in the dorsal raphe (Mlinar et al., 2016), which are not affected by the same temporal smoothing, suggests that the oscillations we record are not likely due to signal aliasing, but instead reflect genuine oscillatory activity. Of course, this does not preclude that other, faster serotonin dynamics are also present in our signal, some of which may be too fast to be observed. For instance, rapid serotonin signaling via the ionotropic 5-HT3a receptors could be missed in our recordings. Additionally, with the fiber photometry approach we adopted, we are limited to capturing spatially broad trends in serotonin levels, potentially overlooking more localized dynamics.

(9) Interactions with other neuromodulators: The study does not explore interactions with other neuromodulators (e.g., norepinephrine, acetylcholine) or their potential ultraslow oscillations.

We agree that the interaction between neuromodulators in the context of ultraslow oscillations is an important issue, which we have addressed in our response to reviewer #1 under ‘Weaknesses.’

(10) Limited exploration of functional significance: While the study suggests a potential role for 5-HT oscillations in memory consolidation and arousal, direct tests of these functional implications are not included.

We agree and reference our answer to (6) regarding memory consolidation. Regarding arousal, direct tests of arousability to different sensory stimuli during different phases of the ultraslow 5-HT oscillation during sleep would be beneficial, in addition to the indirect measures of arousal we examine in the current study, e.g. degree of movement (icEMG) and long range coherence. In line with what we have shown, Cazettes et al. (2021) has demonstrated a direct relationship between 5-HT levels and pupil size, an indicator of arousal level, which like our findings, is consistent across behavioral states.

**Reviewer #3 (Public review):**
Summary:The activity of serotonin (5-HT) releasing neurons as well as 5-HT levels in brain structures targeted by serotonergic axons are known to fluctuate substantially across the animal's sleep/wake cycle, with high 5-HT levels during wakefulness (WAKE), intermediate levels during non-REM sleep (NREM) and very low levels during REM sleep. Recent studies have shown that during NREM, the activity of 5HT neurons in raphe nuclei oscillates at very low frequencies (0.01 - 0.05 Hz) and this ultraslow oscillation is negatively coupled to broadband EEG power. However, how exactly this 5-HT oscillation affects neural activity in downstream structures is unclear.The present study addresses this gap by replicating the observation of the ultraslow oscillation in the 5-HT system, and further observing that hippocampal sharp wave-ripples (SWRs), biomarkers of offline memory processing, occur preferentially in barrages on the falling phase of the 5-HT oscillation during both wakefulness and NREM sleep. In contrast, the raising phase of the 5-HT oscillation is associated with microarousals during NREM and increased muscular activity during WAKE. Finally, the raising 5-HT phase was also found to be associated with increased synchrony between the hippocampus and neocortex. Overall, the study constitutes a valuable contribution to the field by reporting a close association between raising 5-HT and arousal, as well as between falling 5-HT and offline memory processes.Strengths:The study makes compelling use of the state-of-the-art methodology to address its aims: the genetically encoded 5-HT sensor used in the study is ideal for capturing the ultraslow 5-HT dynamics and the novel detection method for SWRs outperforms current state-of-the-art algorithms and will be useful to many scientists in the field. Explicit validation of both of these methods is a particular strength of this study.The analytical methods used in the article are appropriate and are convincingly applied, the use of a general linear mixed model for statistical analysis is a particularly welcome choice as it guards against pseudoreplication while preserving statistical power.Overall, the manuscript makes a strong case for distinct sub-states across WAKE and NREM, associated with different phases of the 5-HT oscillation.Weaknesses:All of the evidence presented in the study is correlational. While the study mostly avoids claims of causality, it would still benefit from establishing whether the 5-HT oscillation has a direct role in the modulation of SWR rate via e.g. optogenetic activation/inactivation of 5-HT axons. As it stands, the possibility that 5-HT levels and SWRs are modulated by the same upstream mechanism cannot be excluded.

We agree that causality claims cannot be made with our data, and acknowledge the interest in exploring causal interactions between ultraslow serotonin oscillations and the correlated activity we measure. We address this point in depth in our answer to Reviewer #2, Weaknesses #3.

**Recommendations for the authors:**

**Reviewer #1 (Recommendations for the authors):**
One major question in the presented data is the nature of the asymmetrical shape of the targeted slow events. How much does it reflect the 5-HT concentration and how much is this shape affected by the dynamics of the designed 5-HT sensor? This needs to be addressed in more detail referencing the original paper for the used sensor.

We have added a paragraph in the Results section of the manuscript to address the asymmetric waveform of the ultraslow 5-HT oscillations and whether it could be affected by the asymmetric kinetics of the GRAB sensor we use: “The waveform of these ultraslow 5-HT oscillations…” (Results, paragraph #4, pg. 5). We include an extended answer to the question here:

Indeed, the GRAB5-HT3.0 sensor we use in the study shows activation response kinetics which are faster than their deactivation time, with time constants at 0.25 s and 1.39 s, respectively (Deng et al. 2024). Likewise, the slope of the rising phase of the ultraslow serotonin oscillation we measure is faster than the slope of the falling phase, and the ratio of time spent in the rising phase versus the falling phase is less than 1, indicating longer falling phases (Figure S2). Although we cannot completely rule out that the asymmetric shape of the ultraslow serotonin oscillations we record is affected by this asymmetry in the 5-HT sensor kinetics, we believe this is unlikely, as the 5-HT signal clearly contains reductions in 5-HT levels that are much faster than the descending phase of the ultraslow oscillation. Although it is difficult to directly compare the different-sized signals, the reported timescales of off kinetics, on the order of a few seconds (Deng et al. 2024), are far below the tens of seconds timescale of the ultraslow oscillation. Furthermore, the finding that some dorsal raphe neurons modulate their firing rate at ultraslow frequencies, and moreover that all examples of such ultraslow oscillations shown display clear asymmetry in rising time versus decay, suggests that the asymmetry we observe in our data could be due to neural activity rather than temporal smoothing by the sensor (Mlinar et al. 2016). In this same direction, another study found similar asymmetry in extracellular 5-HT levels measured with fast scan cyclic voltammetry (FSCV), a technique with greater temporal resolution (sampling rate of 10 Hz) than GRAB sensors, after single pulse stimulation (Bunin and Wightman 1998). In this study, 5-HT was shown to be released extrasynaptically, making the longer clearing time compared to the release time intuitive. Finally, the observation that the onsets and offsets of ripple clusters, recorded with a sampling rate of 20 kHz, are precisely aligned with the peaks and troughs of ultraslow serotonin oscillations (Figure 1, H1-2, columns 2-3) suggests that the duration of the falling phase is not artificially distorted by the temporal smoothing of the sensor dynamics.

Regardless of the dynamics of the serotonin concentration, it should be noted that the elicited neuronal effect might have different dynamics compared to the 5-HT concentration that need to be more studied: to address this one can either examine the average of the broadband LFP (not high passfiltered by the amplifier) or the distribution of simultaneously recorded spiking activity around the peak of ultra-slow oscillations.

We have added Figure S6, showing unit activity relative to the phase of ultraslow serotonin oscillations.

From this analysis, we uncover three groups of units which are largely preserved across states (Figure S6, E vs. F), albeit with a slight temporal shift rightward from NREM to WAKE (Figure S6, C vs. D). Namely, some units spike preferentially during the rising phase, some during the falling phase, and a third group have no clear phase preference. Unit activity during the falling phase is unsurprising, as it is where ripples largely occur, which themselves are associated with spike bursts. During the rising phase, the unit activity we observe could correspond to firing of the hippocampal subpopulation known to be active during NREM interruption states (Jarosiewicz et al. 2002, Miyawaki et al. 2017). While the units’ phase preference was tested based on the category of rising vs. falling phase, as this division described most variation in the data, a few units in the ‘No preference’ group showed heightened activity near the oscillation peak. However, given the very small number of units with this preference, more unit data is needed to describe this group, ideally with high-density recordings. Overall, most units showed a falling vs. rising phase preference, indicating a phase coding of hippocampal activity by 5-HT ultraslow oscillations.

Related to the previous point, it would be helpful to show the average cycle shape of these oscillations (relative to the phase 0 extracted in Figure 3) and do the shape comparison across sessions and also wake/NREM

We agree, and to this end we have added Figure S2. From this waveform analysis, we show that the ultraslow serotonin oscillation is asymmetric, with the rising phase having a greater slope, but shorter length, than the falling phase. While this asymmetry is observed both in NREM and WAKE, the slope difference and length ratio difference in rising vs. falling phase is greater in NREM (Figure S2. B).

In Figure 3D, there seem to be oscillatory rhythms with faster cycles on top of the targeted oscillations. That would make the phase estimation less accurate, e.g. in the left panel, in the second cycle, it is not clear if there are two faster cycles or it is one slow cycle as targeted, and if noted in the rising phase of the second fast cycle there are no ripples. This might suggest that regardless of specific oscillation frequency whenever 5-HT is started to get released, the ripples are suppressed and once the 5-HT is not synaptically effective anymore the ripples start to get generated while the photometry signal starts to wane with the serotonin being cleared. Still, if there is any rhythmicity between bouts of no ripple, it would suggest an ultra-slow regularity in the 5-HT release.

The reviewer is correct to point out that some faster increases in serotonin, which occur on top of the ultraslow oscillations we measure, seem to be associated with decreased ripple incidence, as in the example referenced. The dominance of ultraslow frequencies in the power spectrum of the 5-HT signal suggests, however, that oscillations faster than the ultraslow oscillations we describe are far less prevalent in the data. While there may be some coupling of ripples and other measures to serotonin oscillations of different frequencies, this may be hard or impossible to detect with phase analysis based on their infrequent occurrence and nonstationary nature. In fact, we show in Figure S3 that the strongest phase modulation of ripples by ultraslow serotonin oscillations is observed in the frequencies we use (0.01-0.06 Hz). Methodologically, phase analysis indeed assumes stationary signals, which are rare if not absent in physiological data (Lo et al. 2009), however generally the narrower the frequency band, the better the phase estimation. The narrow frequency band we use provides phase estimates that are largely robust and unaffected by the presence of faster oscillations, as can be seen in the example phase traces shown in Figure 4.

The hypothesis that the rising phase burst of synaptic serotonin is what silences ripples, and that with the clearing of serotonin from the synapses, ripples recover, is a possible explanation of our findings. However, if this were the case, one could expect the ripple rate to increase over the course of the falling phase of ultraslow 5-HT oscillations, as 5-HT decreases, and peak at the trough. This is at odds with what we observe, namely a fairly uniform distribution of ripples along the falling phase (Figure 3F2,F4). Furthermore, the Mlinar et al. 2016 study describes a subpopulation of raphe neurons whose firing rates themselves oscillate at ultraslow frequencies, rather than on-off bursting at ultraslow frequencies, which would argue against this hypothesis. However, as this study looks at a small number of neurons in slices, further in vivo experiments examining firing rates of median raphe neurons are required to understand how the ultraslow oscillation of extracellular serotonin that we measure is generated as well as how it is related to ripple rates.

In Figure 3B, it is not clear why IRI is z-scored. It would be informative to have the actual value of IRI. What is the z relative to? Is it the mean value of IRI in each recording session? Is this to reduce the variability across sessions?

We have now included in Figure 3D a box plot displaying the IRI distributions across different states and sessions. To minimize inter-session variability, data were z-scored within each session for visualization purposes. However, all general linear models were based on raw data, and as a result, the raw differences in IRI are shown in Figure 3C.

Figure 3E, panel labels don't match with the caption

We are grateful to the reviewer for pointing out this mistake, which we have corrected in the updated version of the manuscript.

In the text related to Figure 3E, the related analysis can be more clearly described. "phase preference of individual ripples" does not immediately suggest that the occurring phase of each ripple relative to the targeted oscillation is extracted. I suggest performing this analysis individually for each session and summarizing the results across the sessions.

We have reworded the sentence in Results: 5-HT and ripples to better reflect the analysis performed: “Next, we calculated the ultraslow 5-HT phases at which individual ripples occurred during both NREM and WAKE (3E-F) ...”. Regarding session-level data, we have added Figure S3, which shows session level mean phase vectors, as well as the grand mean across sessions for both NREM and WAKE. Included in this figure are session level means for frequency bands outside of the ultraslow band we used in our study, intended to show that ripples are most strongly timed by the ultraslow band (0.01-0.06 Hz), reflected by the greater amplitude of the mean phase vector for this band.

Figure 3E2, based on the result of ripple-triggered 5-HT in left panels of 2H1-2, one would expect to see a preferred phase closer to 180 (toward the end of the falling phase), it would be helpful to compare and discuss the results of these two analyses.

The reviewer is correct to point out the apparent discrepancy in where the mean ripple falls with respect to the ongoing serotonin oscillation between the two figures mentioned. We have addressed this point in Results: 5-HT and ripples, paragraph #4: “This result appear to be at odds with…”.

Regarding the analysis in 3F, please also compare the power distribution of ripples between NREM and wake. This will help to better understand the potential difference behind the observed difference: how much the strong ripples are comparable between wake and NREM. It is also necessary to report the ripple detection failure rate across ripples with different strengths.

We have added a figure showing analysis done on a subset of the data in which ripples were manually curated in order to evaluate the performance of the ripple detection model (Figure S7) and explanatory text in Methods: Model performance: ‘To ensure that our model …’. In summary, while missed ripples did tend to have lower power than correctly detected ripples, including them did not change the distribution of ripples by the phase of the ultraslow serotonin oscillation (Figure S7C). We would also note that while the phase preference is noisier than what is presented in Figure 3F because this analysis was done with a small subset of all recorded ripples, the fact that ripples occur more clearly on the falling phase is visible for both detected ripples and detected + false negative ripples.

The mixed-effects model examining the influence of 5-HT ultraslow oscillation phase on ripple power revealed no significant effect of state (p = 0.088). This indicates that whether the data were collected during NREM or wake periods did not significantly impact ripple power and that the lack of a significant effect (in Figure 3G,H) in WAKE is probably not due to a difference in the distribution of ripple power between states.

4D, y label is z?

We are grateful for the reviewer to point that out, yes, the y label should be ‘z-score’, as the two traces represent z-scored 5-HT (blue) and z-scored shuffled data (orange). Figure 4D2 and Figure 2H1-2, which show similar data, have been corrected to address this oversight.

Relating to Figure 4, EMG comparison across phases of the oscillations is insightful. Two related and complementary analyses are to compare the theta and gamma power between the falling and rising phases.

We have addressed this suggestion in Figure S5 A-C. While low gamma, high gamma and theta power are modulated identically in NREM, with higher power observed during the falling phase than the rising phase, during WAKE, different patterns can be seen. Specifically, low gamma power shows no phase preference, while high gamma shows a peak near the center of the ultraslow 5-HT oscillation. Theta power, as in NREM, is higher during the falling phase of ultraslow 5-HT oscillations. Increased power across many frequency bands was shown to coincide with decreases in DRN population activity during NREM, which matches with what we report here (Kato et al. 2022). In summary, while NREM patterns are consistent in all frequency bands tested, aligning with the pattern of ripple incidence, in WAKE low and high gamma power show different relationships to ultraslow 5-HT phase.

In the manuscript, we have used the data in both Figure S5 and S6 (unit activity relative to ultraslow 5-HT oscillations), to argue against the idea that our coherence findings result from a lack of activity in the rising phase (see next question), which would have the effect of ‘artificially’ reducing coherence in the falling phase relative the rising phase. The text can be found in Results: 5-HT and hippocampal cortical coherence, paragraph #2.

The results presented in Figure 5 could be puzzling and need to be further discussed: if the ripple band activity is weak during the rising phase, in what circumstances the coherence between cortex and CA1 is specifically very strong in this band?

As mentioned in the previous answer, we have addressed this concern in Results: 5-HT and hippocampal-cortical coherence, paragraph #2. In summary, it is true that the higher coherence in rising phase than in the falling phase for the highest frequency band (termed ‘high frequency oscillation’ (HFO), 100-150 Hz) could be unexpected, given that ripples occur largely during the falling phase. A few points could help explain this finding. Firstly, it should be noted that power in the 100-150 Hz band can arise from physiological activity outside of ripples, such as filtered non-rhythmic spike bursts (Liu et al. 2022), whose coherent occurrence in the rising phase could explain the coherence findings. Secondly, coherence is a compound measure which is affected by both phase consistency and amplitude covariation (Srinath and Ray 2014), thus from only amplitude one cannot predict coherence. Furthermore, HFO power in the cortex is highest near the peak of ultraslow 5-HT oscillations (Figure S5D), as opposed to the falling phase peak in the hippocampus. This shows a lack of covariation in amplitude by phase between the hippocampus and cortex at this frequency band. An alternative explanation of our findings regarding coherence could be that in the rising phase, there is simply little to no activity, which is easier to ‘synchronize’ than bouts of high activity. Hippocampal unit activity in the rising phase (Figure S6) suggests however, that it is not likely to be the absence of activity supporting higher coherence in the rising phase across frequencies. Additional experiments using high density recordings should be conducted to examine 5-HT ultraslow oscillations and their role in gating activity across brain regions, though these results strongly suggest some role exists.

**Reviewer #2 (Recommendations for the authors):**
I would like to offer two comments. I believe that these are not unusual requests, and thus I would like the authors to respond.(1) It would be prudent to investigate the possibility that the observed correlation between ultraslow and hippocampal ripples/microarousals is merely superficial and that there are unidentified confounding factors at play. For example, it would be beneficial to provide evidence that administering a serotonin receptor inhibitor result in the disappearance of the slow oscillation of ripples and microarousals, or that the correlation with ultraslow is no longer present. Please note that the former experiments do not require GRAB5-HT3.0 imaging.

We agree that causality claims cannot be made with our data and acknowledge the interest in exploring causal interactions between ultraslow serotonin oscillations and the correlated activity we measure. We address this point in depth in our answer to Reviewer #2, Weaknesses #3. We would further like to note that given the large number of serotonin receptors and the lack of selectivity of many serotonin receptor antagonists, a pharmacological approach would be difficult, though the results certainly useful. Finally, we highlight the psilocin study, which reported changes in the rhythmic occurrence of microarousals, and therefore likely ultraslow oscillations, after administering a 5-HT2a receptor agonist, suggesting a potential causal effect of 5-HT (via 5-HT2a receptor) on MA occurrence (Thomas et al. 2022).

(2) The slow frequency appears to be associated with the default mode network as observed in fMRI signals. The neural basis of the default mode network remains unclear; therefore, a more detailed examination of this possibility would be beneficial.

We agree that it would be interesting to investigate the role of 5-HT in the neural basis of the DMN.

The DMN as described in humans (Raichle et al. 2001) and rodents (Lu et al. 2012) may indeed include some parts of the hippocampus and perhaps some of our neocortical recordings could also be considered part of the DMN. The fact that the activity across the inter-connected brain structures of the DMN is correlated at ultraslow time scales (Gutierrez-Barragan et al. 2019, Mantini et al. 2007), as well as serotonin’s ability to modulate the DMN is intriguing (Helmbold et al. 2016). Further studies simultaneously recording DMN activity via fMRI and electrical activity via silicon probes, as done in Logothetis et al. 2001, could elucidate further a potential link between ultraslow oscillations and the DMN, with serotonergic modulation as a means to understand any potential contribution of serotonin.

**Reviewer #3 (Recommendations for the authors):**
(1) The impact of the study would benefit from an experiment causally testing the effect of hippocampal 5-HT levels on hippocampal physiology, e.g. using optogenetic manipulations.

We agree that causality claims cannot be made with our data and acknowledge the interest in exploring causal interactions between ultraslow serotonin oscillations and the correlated activity we measure. We address this point in depth in our answer to Reviewer #2, Weaknesses #3.

(2) Data presentation: the figures are of poor resolution, making some diagram details and, more importantly, some example traces (e.g. Figure 1A, right) impossible to see. This should be corrected by either increasing figure resolution or making important figure elements large enough to be readable.

We apologize for the poor resolution and have corrected it in the updated version of the manuscript.

(3) Differences in some figure panels are not statistically assessed: Figure 1H (differences in spectrum peak power), Figure 3E1 & Figure 3E3 (directional bias of the circular distributions), Figure 4C (difference from 0 mean).

We acknowledge this oversight and have added statistical tests for all three figures, as well as further information regarding the models used in Methods: Statistics.

(4) Lines 279-280: the claim that the study shows "organization of activity by ultraslow oscillations of 5-HT" implies a causal role of 5-HT in organizing hippocampal activity. I suggest that this statement be toned down to reflect the correlational nature of the presented evidence.

We have rephrased the sentence in question to the following: “In our study, including both NREM and WAKE periods allowed us to additionally show that the temporal organization of activity relative to ultraslow 5-HT oscillations operates according to the same principles in both states...”, which we believe better reflects the temporal correlation we describe.

(5) While the study claims to use the EMG (i.e. electromyograph) signal, it does not describe any electrodes placed inside the muscle in the methods section. The SleepScoreMaster toolbox used in the study estimates the EMG using high-frequency activity correlated across recording channels, so I assume this is how this signal was obtained. While such activity may well reflect muscular noise to some degree, it is an indirect measure as the electrodes are not in the muscle. Since the EMG signal is central to the message of the manuscript, the method for calculating it should be described in the methods section and it should be explicitly labelled as an indirect measure in the main text, e.g. by referring to this signal as pseudo-EMG.

We agree and have added explanatory text to the State Scoring subsection in Methods. Given that the EMG we refer to is derived from intracranial data, and not from traditional EMG probes, we now refer to the EMG as intracranial EMG, or icEMG for short, throughout the main text.

(6) Is ripple frequency or ripple duration different across the rising and falling phases of the ultraslow oscillation?

We have now investigated this suggestion in Figure S4, where we show that ripple frequency is higher in the falling phase than rising phase, while ripple duration appears to show no phase preference.

(7) Lines 315-317: I am not sure why the manuscript refers to the coupling between EMG and 5-HT levels as 'puzzling' given that, as stated, the locomotion-inducing effects of 5-HT are well documented. While the fact that even non-locomotory motor activity may be associated with 5-HT rise is certainly interesting (although not sure if 'puzzling'), the manuscript does not directly compare the association of 5-HT levels with locomotory and non-locomotory EMG spikes. Thus, I think this discussion point is not fully warranted.

We agree and have rephrased the discussion point in question to reflect that the EMG link to serotonin oscillations is not necessarily surprising, given both the literature linking 5-HT and spontaneous movement in the hippocampus, as well as the involvement of 5-HT in repetitive movements, where the role for a regularly-occurring oscillation is perhaps more intuitive.

(8) Line 441: Reference #67 does not describe the use of fiber photometry.

The reviewer is to correct to point out this typo, which has been now corrected. The reference in question should be 64, where fiber photometry experiments are described. For further clarity, we have changed our referencing scheme to include authors and years in in-text references.

(9) In Figures 3E1-3, the phase has different bounds than in the other Figures in the manuscript (0:360 vs -180:180), this should be corrected for consistency.

We agree and have made changes so that all figures have a phase range of -180 to 180°.

References

(1) Z. M Aghajan, G. Kreiman, I. Fried, Minute-scale periodicity of neuronal firing in the human entorhinal cortex. Cell Rep 42, 113271 (2023).

(2) M.A. Bunin, R.M. Wightman (1998). Quantitative Evaluation of 5-Hydroxytryptamine (Serotonin) Neuronal Release and Uptake: An Investigation of Extrasynaptic Transmission. J. Neurosci. 18 (13) 4854-4860

(3) F. Cazettes, D. Reato, J. P. Morais, A. Renart, Z. F. Mainen, Phasic Activation of Dorsal Raphe Serotonergic Neurons Increases Pupil Size. Curr Biol 31, 192-197.e4 (2021).

(4) Cole SR, Voytek B. Brain Oscillations and the Importance of Waveform Shape. Trends Cogn Sci. 21(2):137-149 (2017).

(5) F. Deng, et al., Improved green and red GRAB sensors for monitoring spatiotemporal serotonin release in vivo. Nat Methods 21, 692–702 (2024).

(6) C. Dong, et al., Psychedelic-inspired drug discovery using an engineered biosensor. Cell 184, 2779-2792.e18 (2021).

(7) A. Eban-Rothschild, L. Appelbaum, L. de Lecea, Neuronal Mechanisms for Sleep/Wake Regulation and Modulatory Drive. Neuropsychopharmacol. 43, 937–952 (2018).

(8) M. S. Fanselow, H.-W. Dong, Are the dorsal and ventral hippocampus functionally distinct structures? Neuron 65, 7–19 (2010).

(9) J. S. Farrell, E. Hwaun, B. Dudok, I. Soltesz, Neural and behavioural state switching during hippocampal dentate spikes. Nature 1–6 (2024). https://doi.org/10.1038/s41586-024-07192-8.

(10) De Filippo, R., & Schmitz, D. (2024). Transcriptomic mapping of the 5-HT receptor landscape. Patterns (New York, N.Y.), 5(10), 101048.

(11) M. J. Fisher, et al., Neural mechanisms of sensory gating: Insights from human and animal studies. NeuroImage 207, 116374 (2020).

(12) P. Fries, D. Nikolić, W. Singer, The gamma cycle. Trends in Neurosciences 30, 309–316 (2007).

(13) S. Gonzalo Cogno, et al., Minute-scale oscillatory sequences in medial entorhinal cortex. Nature 625, 338–344 (2024).

(14) D. Gutierrez-Barragan, M. A. Basson, S. Panzeri, A. Gozzi, Infraslow State Fluctuations Govern Spontaneous fMRI Network Dynamics. Current Biology 29, 2295-2306.e5 (2019).

(15) K. Helmbold, et al., Serotonergic modulation of resting state default mode network connectivity in healthy women. Amino Acids 48, 1109–1120 (2016).

(16) B. Jarosiewicz, B. L. McNaughton, W. E. Skaggs, Hippocampal Population Activity during the Small-Amplitude Irregular Activity State in the Rat. J. Neurosci. 22, 1373–1384 (2002).

(17) J. T. Jordan, The rodent hippocampus as a bilateral structure: A review of hemispheric lateralization. Hippocampus 30, 278–292 (2020).

(18) T. Kato, et al., Oscillatory Population-Level Activity of Dorsal Raphe Serotonergic Neurons Is Inscribed in Sleep Structure. J. Neurosci. 42, 7244–7255 (2022).

(19) M.A. Kim, H. S. Lee, B. Y. Lee, B. D. Waterhouse, Reciprocal connections between subdivisions of the dorsal raphe and the nuclear core of the locus coeruleus in the rat. Brain Research 1026, 56–67 (2004).

(20) C. Kjaerby, et al., Memory-enhancing properties of sleep depend on the oscillatory amplitude of norepinephrine. Nat Neurosci 25, 1059–1070 (2022).

(21) S. Lecci, et al., Coordinated infraslow neural and cardiac oscillations mark fragility and offline periods in mammalian sleep. Sci Adv 3, e1602026 (2017).

(22) A. A. Liu, et al., A consensus statement on detection of hippocampal sharp wave ripples and differentiation from other fast oscillations. Nat Commun 13, 6000 (2022).

(23) M.-T. Lo, P.-H. Tsai, P.-F. Lin, C. Lin, Y. L. Hsin, The nonlinear and nonstationary properties in eeg signals: probing the complex fluctuations by hilbert–huang transform. Adv. Adapt. Data Anal. 01, 461–482 (2009).

(24) N. K. Logothetis, J. Pauls, M. Augath, T. Trinath, A. Oeltermann, Neurophysiological investigation of the basis of the fMRI signal. Nature 412, 150–157 (2001).

(25) H. Lu, et al., Rat brains also have a default mode network. Proc Natl Acad Sci U S A 109, 3979–3984 (2012).

(26) D. Mantini, M. G. Perrucci, C. Del Gratta, G. L. Romani, M. Corbetta, Electrophysiological signatures of resting state networks in the human brain. Proc Natl Acad Sci U S A 104, 13170– 13175 (2007).

(27) J. E. Markowitz, et al., The striatum organizes 3D behavior via moment-to-moment action selection. Cell 174, 44-58.e17 (2018).

(28) H. Miyawaki, Y. N. Billeh, K. Diba, Low Activity Microstates During Sleep. Sleep 40, zsx066 (2017).

(29) B. Mlinar, A. Montalbano, L. Piszczek, C. Gross, R. Corradetti, Firing Properties of GeneticallyIdentified Dorsal Raphe Serotonergic Neurons in Brain Slices. Front Cell Neurosci 10, 195 (2016).

(30) A. Osorio-Forero, et al., Noradrenergic circuit control of non-REM sleep substates. Current Biology 31, 5009-5023.e7 (2021).

(31) S. Panzeri, N. Brunel, N. K. Logothetis, C. Kayser, Sensory neural codes using multiplexed temporal scales. Trends in Neurosciences 33, 111–120 (2010).

(32) M. E. Raichle, et al., A default mode of brain function. Proc Natl Acad Sci U S A 98, 676–682 (2001).

(33) R. Srinath, S. Ray, Effect of amplitude correlations on coherence in the local field potential. J Neurophysiol 112, 741–751 (2014).

(34) B. P. Staresina, J. Niediek, V. Borger, R. Surges, F. Mormann, How coupled slow oscillations, spindles and ripples coordinate neuronal processing and communication during human sleep. Nat Neurosci 26, 1429–1437 (2023).

(35) C. W. Thomas, et al., Psilocin acutely alters sleep-wake architecture and cortical brain activity in laboratory mice. Transl Psychiatry 12, 77 (2022).

(36) G. F. Turi, et al., Serotonin modulates infraslow oscillation in the dentate gyrus during Non-REM sleep. eLife 13 (2025).

(37) J. Vazquez, H. A. Baghdoyan, Basal forebrain acetylcholine release during REM sleep is significantly greater than during waking. American Journal of Physiology-Regulatory, Integrative and Comparative Physiology 280, R598–R601 (2001).

(38) J. Wan, et al., A genetically encoded sensor for measuring serotonin dynamics. Nat Neurosci 24, 746–752 (2021).

(39) Y. Zhang, et al., Cholinergic suppression of hippocampal sharp-wave ripples impairs working memory. Proc. Natl. Acad. Sci. U.S.A. 118, e2016432118 (2021).

(40) Y. Zhang, et al., Interaction of acetylcholine and oxytocin neuromodulation in the hippocampus. Neuron (2024).